# C1q drives neural stem cell quiescence by regulating cell cycle and metabolism through BAI1

Katja M. Piltti [1,2,3,4] ✉, Anita Lakatos [1,11], Francisca Benavente-Perez[1,11], Zeina H. Elrachid [1], Aileen A. Nava [1,9], Nathaniel T. Addonizio [1], Victoria Nguyen [5], Atena Zahedi [1], Wei A. Song [6,10], Xiyu Chen [1], Josh David [1], Andreina Portillo[6], Ayla O. Manughian-Peter[5], Andrea N. Rodriguez[7], Andrea Anzalone[1], Catherine M. Cahill [8] & Aileen J. Anderson [1,2,3,4] ✉

C1q levels in the CNS are elevated by inflammation and neurovascular trauma, yet the consequence of this increase for neural stem cell (NSC) regeneration response remain unknown. We have recently identified C1q receptor candidates that regulate NSC behavior. One of these is Brain Angiogenesis Inhibitor 1 (BAI1, ADGRB1), which has no previously discovered role in NSC. Here, we show that C1q acts in a BAI1-dependent manner to modulate NSC quiescence via two parallel mechanisms. First, negative regulation of MDM2, driving cell cycle suppression through p53. Second, endocytic internalization of C1q-BAI1-complex, driving regulation of p32 (C1qBP) and metabolic reprogramming towards aerobic glycolysis. We validated the biological significance of BAI1 in a male hNSC line in vivo using a female mouse model of acute spinal cord injury (SCI). These findings are relevant for a multiplicity of CNS disorders, and illuminate complex connections between C1q, cell cycle, and metabolism. Together, these data provide valuable insight into C1q-mediated regulation of NSC transition between activation and quiescence, processes fundamental for tissue development and repair.

Adult tissue-specific stem cells maintain tissue homeostasis by shifting from quiescence to activation. Strict regulation over cell activation, cell cycle progression, and lineage-specific differentiation is crucial to prevent stem cell pool exhaustion[1]. During aging, tissues exhibit progressive declines in stem cell activation, contributing to loss of regenerative capacity and pathology[2]. This is exacerbated in the CNS, where regenerative capacity is already limited relative to other tissues[3–5]. Additionally, CNS aging, neurodegenerative disease, and injury are associated with prolonged inflammation[6–9], including exposure of the NSC-neurovascular niche to complement C1q derived from microglia[10,11] and blood-brain/spinal cord-barrier (BBB/BSB) dysfunction[12,13]. This results in a 400-fold increase in the accumulation

[1]Sue and Bill Gross Stem Cell Research Center, University of California Irvine, Irvine, CA, USA. [2]Institute for Memory Impairments and Neurological Disorders, University of California Irvine, Irvine, CA, USA. [3]Department of Anatomy and Neurobiology, University of California Irvine, Irvine, CA, USA. [4]Department of Physical Medicine and Rehabilitation, University of California Irvine, Irvine, CA, USA. [5]California State University Long Beach, Long Beach, CA, USA. [6]California State University Fullerton, Fullerton, CA, USA. [7]California State University San Bernardino, San Bernardino, CA, USA. [8]Department of Psychiatry and Biobehavioral Sciences, Semel Institute for Neuroscience & Human Behavior, University of California Los Angeles, Los Angeles, CA, USA. [9]Present address: Department of Human Genetics, University of California Los Angeles, Los Angeles, CA, USA. [10]Present address: Department of Biological Chemistry, University of California Irvine, Irvine, CA, USA. [11]These authors contributed equally: Anita Lakatos, Francisca Benavente-Perez. ✉e-mail: kpiltti@uci.edu; aja@uci.edu

of C1q in the aged brain[6]. Similarly, C1q levels in peripheral serum increase almost twofold during aging[14]. Age-related increases in C1q induce declines in skeletal muscle regeneration via direct effect on satellite stem cell proliferation[15], however, the effect of C1q on declines in NSC activation (proliferation) vs. maintenance (self-renewal) has not been investigated.

Mitochondria and metabolic rewiring have emerged as central regulators of activation and maintenance of the NSC pool[16,17], and both CNS aging and neurodegenerative disease are linked to impairments in energy metabolism and metabolic dysfunction. Although complement proteins regulate immune cell metabolism and function via the complosome[18,19], a direct role or mechanism of action for C1q in neural metabolism has not been established.

We recently identified C1q candidate cell surface receptors in NSC[20], enabling the application of gene editing to study the role of C1q in NSC function. Here, we demonstrate that blood plasma C1q concentrations associated with aging and BBB/BSB breakdown drive declines in NSC proliferation in vitro and in vivo in a CNS trauma model. We identify that the underlying mechanism is dependent on an interaction with BAI1, and distinct from that shown for muscle satellite cells, driving not only cell cycle arrest but also metabolic changes associated with stem cell quiescence.

## Results

### C1q promotes hNSC quiescence

To investigate the effect of C1q on hNSC proliferation, we exposed hNSC to purified human C1q at physiological concentrations ranging from 0.1 nM to 300 nM, followed by a BrdU incorporation assay. As reported for muscle satellite cells, C1q concentrations associated with circulating blood plasma [100nM-300nM] dramatically decreased the proliferating hNSC population (Fig. 1a, b). hNSC survival (RealTime-Glo Annexin V timecourse analysis; Fig. 1c, d) and apoptosis (cleaved caspase 3, CC3; Supplementary Fig. 1a, b) were unaltered in C1q-treated hNSC vs. non-treated controls. In addition, C1q-treated hNSC exhibited a rebound in proliferation when returned to normal in vitro growth conditions (Fig. 1e), demonstrating reversibility of this effect. The effect of C1q on hNSC proliferation was blocked by co-treatment with a previously characterized C1q-neutralizing antibody but not an IgG control (nAb; Fig. 1f)[5,20].

We tested the role of cell cycle modulation in C1q-driven proliferation decline using Fucci-mNSC (Fig. 1g–i, Supplementary Fig. 1c)[21–23]. Live-cell time-lapse and quantification of cell cycle dynamics revealed that C1q [200 nM] promoted G0/G1 phase arrest and decreased progression to G1/S or S/G2/M vs. non-treated controls (Fig. 1h, i). Consistent with RealTime-Glo Annexin V analysis, C1q-driven G0/G1 arrest increased cell survival rather than cell death in Fucci-mNSC (Fig. 1j and Supplementary Fig. 1d, e). These data identify that exogenous C1q negatively regulates cell cycle progression/proliferation in NSC.

A reversible loss of proliferation combined with cell cycle arrest at G0 in the absence of cell death (Fig. 1a–j) is consistent with stem cell quiescence[24,25], which would predict a decrease in self-renewal. As hypothesized, clonal neurosphere assays revealed a significant decrease in hNSC self-renewal during C1q [200 nM] exposure vs. non-treated controls (Fig. 1k, l). No significant change in neurosphere size was detected during C1q [200 nM] treatment vs. non-treated controls (Supplementary Fig. 1f). Additionally, stem cell quiescence is distinct from cell cycle arrest during terminal differentiation. Analysis of total protein lysates from hNSC exposed to C1q [200 nM] during in vitro differentiation revealed significant increases in undifferentiated NSC/progenitor markers, nestin and GFAPδ[26], but not lineage commitment markers (Fig. 1m). RNAseq further identified significant transcriptomic enrichment of quiescence genes in C1q [200 nM] treated hNSC (Fig. 1n, grey), shown by radar plots identifying enriched pathways (labels) and magnitude of change (distance from center). These data indicate that C1q induces quiescence in hNSC.

While the effect of C1q on NSC proliferation was reversible after short-term exposure (up to 48 h, Fig. 1e), CNS injury/aging can cause prolonged C1q elevation. We therefore tested whether long-term exposure to C1q has a priming effect on hNSC proliferation or self-renewal. hNSC neurosphere cultures were exposed to C1q for 2, 4, 6, or 8 weeks, at which time C1q was withdrawn and cells returned to normal in vitro growth conditions for analysis 5 weeks later. Consistent with short-term exposure effects (Fig. 1e), hNSC proliferation declines were reversed after C1q withdrawal (Fig. 1o). hNSC self-renewal declines (Fig. 1l), however, persisted after long-term priming (Fig. 1p), and were exacerbated by the length of exposure. Long-term C1q [200 nM] priming caused no significant differences in neurosphere size relative to non-treated controls (Supplementary Fig. 1g), supporting the interpretation that there were fewer neurosphere-initiating cells present in the C1q [200 nM] primed group over time. Apoptotic cell death again remained unchanged (Supplementary Fig. 1h, i), and hNSC retained multi-lineage differentiation capacity (Supplementary Fig. 1j). Depending on factors within the niche[24,25], quiescence exit of NSC can occur through any of the following modes of cell divisions: self-renewing symmetric, in which two new NSC; self-renewing asymmetric are produced, in which one of the two cells remains as an NSC and another initiates terminal differentiation; or self-consuming symmetric, in which two faster proliferating transit amplifying (TA) progenitors that both initiate terminal differentiation are produced[27]. The self-consuming symmetric cell divisions leading to terminal differentiation can thus deplete the originating stem cell population and result in stem cell exhaustion. Therefore, the observed long-term C1q priming induced proliferation rebound and increase, coupled with an overall decline in self-renewal, could be interpreted as NSC shifting their mode of division from self-renewing to self-consuming. This would suggest that extended exposure to C1q at blood plasma concentrations could affect maintenance of the NSC pool and long-term tissue turnover, providing an important insight into how C1q could contribute to CNS pathological processes and impact repair capacity. Critically, decline of the neural stem cell pool is posited to be a major factor in the age-related decline of regenerative functions in the CNS, and is thought to derive from increased quiescence and terminal differentiation[24,28].

C1q-driven proliferation decline has been previously reported in muscle satellite cells, where C1q in C1 complex binds to Frizzled receptor, resulting in C1s-mediated cleavage of LPR6 and Wnt pathway activation[15]. To investigate whether C1q modulates hNSC proliferation through the same mechanism, we evaluated expression of these transcripts. hNSC expressed Wnt ligands, Frizzed receptors, LRP, including LRP 5/6, and C1s/C1r, but not C1q or downstream complement pathway components required for formation of the membrane attack complex (Supplementary Table 1). Despite the absence of C1q transcripts, paracrine/exogenous C1q could theoretically combine with autocrine C1r/C1s to form the C1 complex. However, in contrast to muscle satellite cells, neither a C1s inhibitor (Fig. 1q) nor a C1 complex-inhibitor (Fig. 1r) rescued C1q-induced proliferation decline. Further, purified human C1 complex had no effect on hNSC proliferation (Fig. 1s). These data point to a different mechanism for the action of C1q on NSC, in which C1q selectively acts extracellularly as a single ligand.

### C1q modulates p53 activity towards quiescence

C1q has been shown to drive activation of specific intracellular signaling pathways in a concentration-dependent manner in hNSC[20]. We searched these data for an intracellular signaling signature that could provide mechanistic insight, identifying that C1q [200 nM] uniquely increases phosphorylation of p53 serine 15 (p53$^{Ser15}$) (Fig. 2a). We confirmed this observation by Western blot vs. non-treated and temperature-inactivated C1q controls (T̊ Inact; Fig. 2b). p53 is a critical regulator of stem cell proliferation and self-renewal[29], and p53$^{Ser15}$ is important for protein stability and transcriptional activation, particularly of the cell cycle arrest protein p21[30,31]. Nuclear p53 functions as a

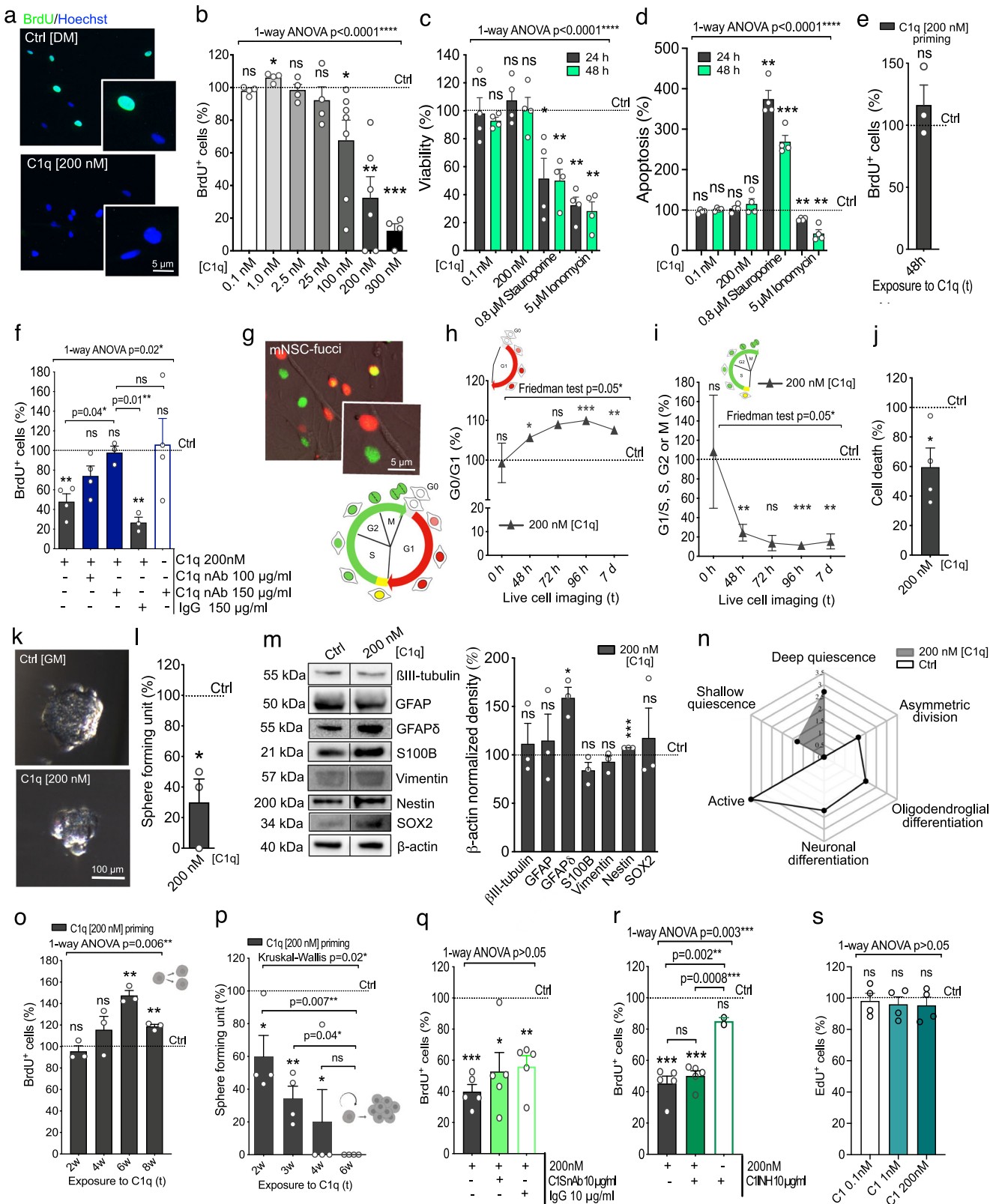

transcription factor; however, mitochondria/cytoplasmic p53 is linked to other cellular functions[32–34]. We analyzed the effect of C1q on phospho-p53$^{Ser15}$ subcellular localization, identifying a selective increase in nuclear localization in C1q [200 nM]-treated hNSC vs. controls (Fig. 2c, d). Consistent with nuclear p53 localization, genes associated with p53 downstream activation (Fig. 2e, f) and cell cycle arrest (Fig. 2g) were significantly enriched and differentially expressed

in C1q [200 nM]-treated vs. control hNSC in RNAseq. These data identify p53 as a possible driver for the C1q effect on NSC cell cycle.

## C1q-driven increase in p53 signal transduction is BAI1-dependent

p53 sub-cellular localization and function are modulated by the MDM2 E3 ubiquitin ligase[35]. In medulloblastoma, the G-protein-coupled

**Fig. 1 | C1q promotes NSC quiescence via a C1s-independent mechanism.**
**a** Representative BrdU immunostaining in non-treated vs. C1q-treated hNSC.
**b** BrdU incorporation in hNSC treated with C1q [0.1–300 nM][5,14,78,113–115] relative to
non-treated controls (dashed line; $n = 4$, 6, or 7 as indicated). **c, d** Cell viability (**c**)
and apoptosis (**d**) in C1q-treated hNSC relative to controls ($n = 4$). Positive controls
staurosporine or ionomycin. **e** BrdU analysis after 48 h C1q-preconditioning vs.
controls ($n = 3$). **f** BrdU analysis in hNSC co-treated with C1q and C1q-neutralizing
antibody (nAb) or IgG vs. control ($n = 3$ or 4 as indicated). **g** Representative image of
Fucci-mNSC[23] and diagram[#]. **h, i** Quantification of cell cycle via live-cell imaging:
G0/G1 (non-fluorescent/Cdt1-red; **h**) and S/G2/M (Geminin-green; **i**) in C1q-treated
mNSC vs. controls ($n = 4$)[#]. **j** Cell survival analysis in C1q-treated Fucci-mNSC over
7 days via single-cell tracking ($n = 4$). **k, l** Example images of neurospheres (**k**) and
self-renewal capacity during C1q exposure vs. controls (**l**) ($n = 3$). **m** Western blot of
lineage-specific markers in C1q-treated hNSC normalized to β-actin and non-treated
controls ($n = 3$). Black lines, non-adjacent lanes. **n** Pathway enrichment analysis of
quiescence, activation, and neural differentiation associated transcripts in RNA-seq
(FDR $p \le 0.05$, $n = 3$). **o, p** Long-term priming effect of C1q on hNSC proliferation
(**o**, $n = 3$) and self-renewal (**p**, $n = 4$). **q–s** BrdU/EdU analysis in hNSC co-treated with
C1q and C1s nAb (**q**, $n = 5$), C1 inhibitor (**r**, $n = 4$), or C1 complex (**s**, $n = 4$). Mean ±
s.e.m., **b–d**, **o**, **q–s** 1-way ANOVA test and 1-sample 2-tailed t-test, **f** 1-way ANOVA test
with post hoc and 1-sample 2-tailed t-test, **e**, **j**, **l**, **m** 1-sample 2-tailed t-test,
**h**, **i** Nonparametric Friedman test and 1-sample 2-tailed t-test or Wilcoxon signed
2-tailed rank test, **n** two-sided permutation-based enrichment score test with
Benjamini−Hochberg correction for multiple comparisons, **p** Kruskal-Wallis test
with post hoc and 1-sample 2-tailed t-test. $n$ = biologically independent experi-
ments. ns = not significant; *$p \le 0.05$; **$p \le 0.01$; ***$p \le 0.001$; ****$p \le 0.0001$. Exact p-
values, $n$, full blots with each β-actin available in Source Data. [#]Illustrations created
in BioRender. Piltti, K. (2025) https://BioRender.com/rvswasg.

---

receptor BAI1 binds MDM2, protecting p53 from degradation and
suppressing tumor cell proliferation[36]. We recently identified C1q as a
ligand for BAI1 in NSC[20]. While BAI1 engagement with its ligands has
been shown to modulate intracellular signaling[37,38], whether BAI1
engagement with ligands such as C1q can alter BAI1-MDM2 binding to
modulate protein degradation is unknown. We hypothesized that C1q-
BAI1 engagement could increase BAI1-MDM2 binding as a mechanism
for C1q-driven increases in p53 phosphorylation and nuclear translo-
cation (Fig. 3a). We generated a BAI1 knockout (KO) hNSC line to test
this hypothesis. BAI1 KO cells retained normal characteristics, includ-
ing karyotype, CD133+ stem cell proportion, stable growth rate under
in vitro growth conditions, migration response, and multipotency in
neural lineage differentiation (Supplementary Fig. 2). We validated loss
of BAI1 in KO hNSC by Western blot (Fig. 3b), as well as loss of C1q-BAI1
interaction in these cells by proximity ligation assay (PLA; Fig. 3c, d)
using commercial BAI1 antibodies tested in BAI1 HEK293T over-
expression vs. negative control lysates (Supplementary Fig. 3a). We
investigated whether C1q altered BAI1-MDM2 binding in wild type (WT)
hNSC by PLA, identifying a strongly significant increase in response to
C1q that was abolished in BAI1 KO hNSC (Fig. 3e, f). PLA specificity was
validated using a mismatched (negative) interaction control (MET-
MDM2; Fig. 3g, h). We also evaluated the dependence of C1q-mediated
changes on p53 phosphorylation on BAI1. We replicated the C1q-
mediated increase in p53 normalized total p53[Ser15] (Fig. 3i) and shift in
p53[Ser15] localization to the nucleus (Fig. 3j, k) in BAI1 WT hNSC, and
confirmed that these effects were reversed by BAI1 KO. These data
demonstrate that C1q-BAI1 interaction promotes BAI1-MDM2 binding
and p53 activity.

Critically, BAI family proteins have other known roles in cellular
function, suggesting that MDM2/p53 signaling may not be the only
mechanism modulating hNSC quiescence. Ligand-receptor complex
internalization by endocytosis is key to GPCR modulation of intracel-
lular signaling[39,40]. Both BAI1 and BAI3 GPCR family members have
identified roles in endocytosis[41–46]. In parallel, intracellular comple-
ment is emerging as an orchestrator of cellular functions in immune
cells[47]; however, only two reports have described intracellular C1q in
neural lineage cells[48,49], and no intracellular role for C1q has been
established in CNS cells. We therefore hypothesized that C1q could be
internalized in complex with BAI1 via endocytosis, modulating the
proliferative state of hNSC intracellularly through a second
mechanism.

## BAI1-C1q endocytosis decreases total p32

C1q internalization kinetics were analyzed using single-cell Image-
stream multispectral imaging flow cytometry[50]. C1q immunolabeling
was detected in association with 46% of in-focus single cells in C1q
[200 nM]-treated cultures (Fig. 4a); component masking analysis[51] of
this subpopulation identified C1q internalization within 5 min,
increasing through 24 h post-treatment (Fig. 4b, c, e, Supplementary
Figs. 3b, 4m). Internalized C1q colocalized with early endosomes

(Fig. 4d, f), but not with Golgi (Fig. 4g). We further verified C1q inter-
nalization in hNSC using immunogold electron microscopy (Fig. 4h).

We investigated the intracellular proteins/pathways with which
internalized C1q interacts via an unbiased forward screen using
nanoLC-MS/MS, and a pull-down approach with a cell-permeable
cross-linker in hNSC monolayer cultures. While this approach limits
sensitivity, the 3D structure of C1q-bait and interacting intracellular
prey-proteins is maintained, maximizing biological relevance. We
detected a total of 214 prey proteins that were significantly different in
C1q vs. non-treated samples. Subcellular enrichment analysis identified
a significant association of these proteins with several specific com-
partments involved in receptor-mediated endocytic trafficking of C1q
(Fig. 4i, Supplementary Table 2), including the cytoskeletal, cyto-
plasmic, vesicular, lysosome, ribosome and endoplasmic reticular
compartments. Subcellular interactions were specific, as no enrich-
ment was detected in the mitochondria or peroxisome, which are not
linked to endocytic pathways of protein internalization. Further sup-
porting specificity, the cell-permeant cross-linker paradigm did not
detect enrichment at the cell membrane.

We next determined whether C1q is internalized in complex with
BAI1. Imagestream analysis was validated using two matched positive
controls (Fig. 4j and Supplementary Fig. 4a–i), and a mismatched
negative control (Fig. 4k)[52,53]. In C1q [200 nM]-treated cultures,
BAI1 was detected in 78% of in-focus single cells (Fig. 4l, n), and
intracellular C1q-BAI1 co-localization was identified within 5 min
(Fig. 4o, Supplementary Fig. 3c, Supplementary Fig. 4j, k). BAI1 KO
decreased both the proportion of in-focus single cells with intracellular
C1q (Fig. 4p and Supplementary Fig. 4l) and the overall quantity of
internalized C1q per cell within this sub-population (Fig. 4q and Sup-
plementary Fig. 4n)[54]. These data suggest that C1q-BAI1 engagement
not only modulates p53 signaling but also serves as an endocytic portal
for C1q in NSC.

In T cells, C1q interaction with cell surface p32 results in C1q
endocytosis followed by C1q-p32 transport to the mitochondria via an
unknown mechanism[55]. In contrast to T-cells, no evidence of cell sur-
face p32 was detected in hNSC by either Imagestream (Supplementary
Fig. 4o) or protein enrichment of specific cellular compartments (Fig.
4r, s). Rather, p32 was localized intracellularly, particularly in the
mitochondrial matrix (Supplementary Fig. 4p), consistent with a well-
established role for p32 in mitochondrial protein translation and
OXPHOS activity[56,57]. We therefore hypothesized that BAI1-mediated
C1q endocytosis would enable intracellular C1q-p32 interaction. In C1q
[200 nM]-treated cultures, p32 was detected in 82% of in-focus single
hNSC (Fig. 4m, t), and intracellular C1q-p32 co-localization was iden-
tified within 30 min (Fig. 4u). Intracellular C1q-p32 interaction was
confirmed by PLA (Fig. 4v, w). Because Parkin and ArgII have been
shown to regulate p32 stability[58], we hypothesized that C1q might do
the same. Consistent with this idea, p32 protein was significantly
decreased post-C1q treatment (Fig. 4x). This decrease was dependent
on BAI1, as shown by Imagestream (Fig. 5a, b), PLA (Fig. 5c, d), total p32

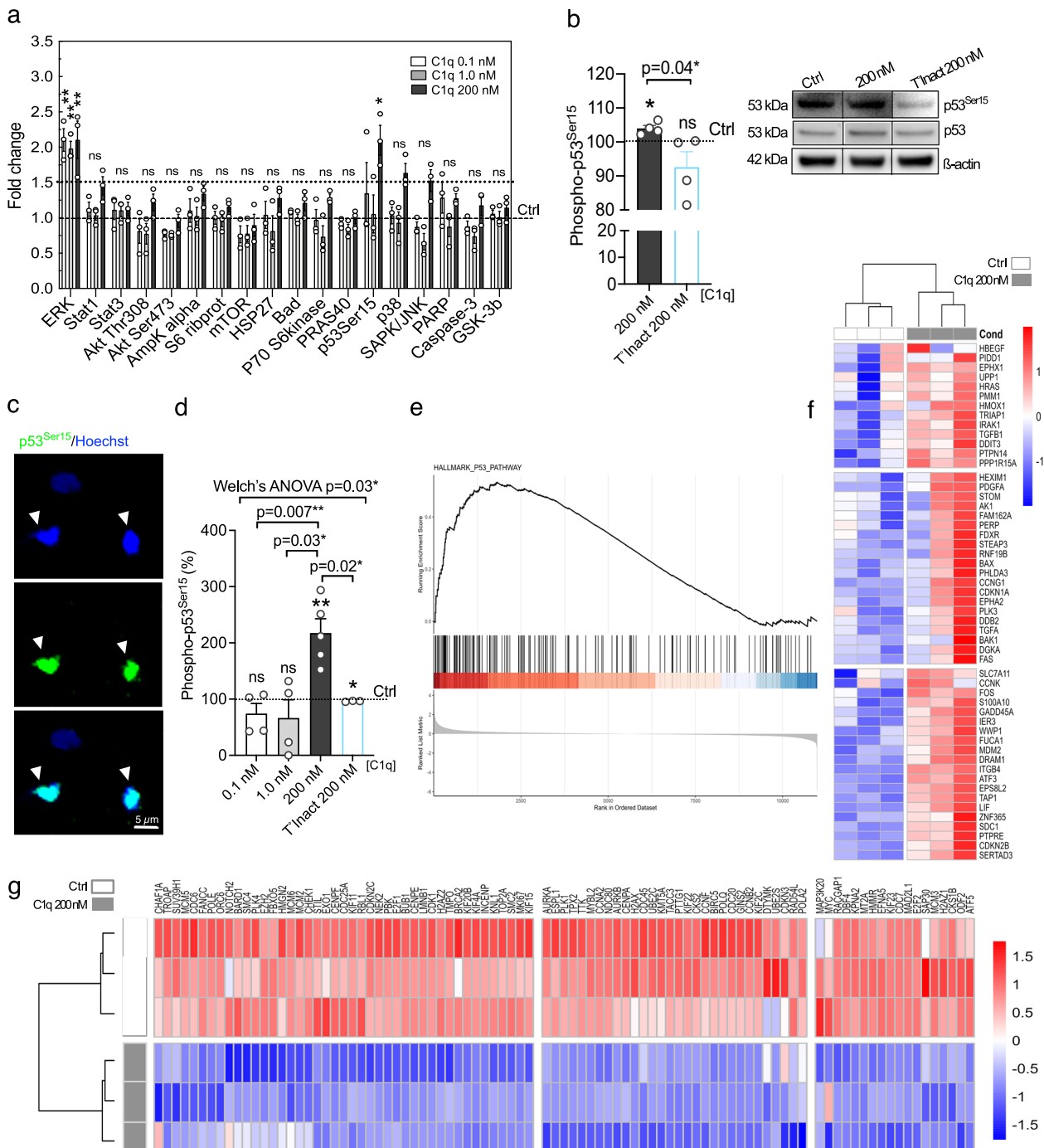

**Fig. 2 | Blood plasma C1q concentration activates p53 signal transduction driving cell cycle quiescence. a** Phosphoarray analyses[20] demonstrate selective increase in phosphorylation of p53[Ser15] in C1q [200 nM]-treated hNSC. Increase in pErk1/2 phosphorylation was not specific to C1q [200 nM] and was detected in all tested C1q concentrations. Quantified optical intensity is shown relative to control (ctrl, dashed line); ≥1.5-fold change was set for statistical significance ($n = 3$). **b** Western blot analysis of phospho-p53[Ser15] in hNSC treated with C1q [200 nM] or heat-inactivated C1q [200 nM] (T°Inact 200 nM) normalized to pan-p53 and non-treated controls ($n = 4$). Black line non-adjacent lanes; full blots available in the Source Data. **c** Representative image of phospho-p53[Ser15] nuclear localization in hNSC after immunostaining. **d** Quantification of phospho-p53[Ser15] nuclear localization in hNSC treated with C1q [200 nM] or T°Inact C1q [200 nM] relative to non-treated controls ($n = 3, 4$, or $5$ as indicated). **e–g** C1q [200 nM] effect on transcriptional activation of p53 downstream targets (**e**, **f**) and proliferation arrest/cell cycle arrest associated genes (**g**) (RNA-seq, $n = 3$). GSEA plot enrichment of HALL-MARK_p53 pathway in (**e**) (FDR $p ≤ 0.05$) and heat map of the contributing transcripts in (**f**). Heat map of cell cycle transcripts in (**g**). Red up-regulation, blue down-regulation. Mean ± s.e.m., **a** 1-sample 2-tailed t-test, **b** unpaired 1-tailed Welch's t-test and 1-sample 2-tailed t-test, **d** Unequal variance Welch's ANOVA with post hoc test and 1-sample 2-tailed t-test, **e** Two-sided permutation-based enrichment score test with Benjamini–Hochberg correction for multiple comparisons. $n =$ biologically independent experiments. ns not significant; *$p ≤ 0.05$; **$p ≤ 0.01$; ***$p ≤ 0.001$; ****$p ≤ 0.0001$. Exact $p$-values and $n$ available in Source Data.

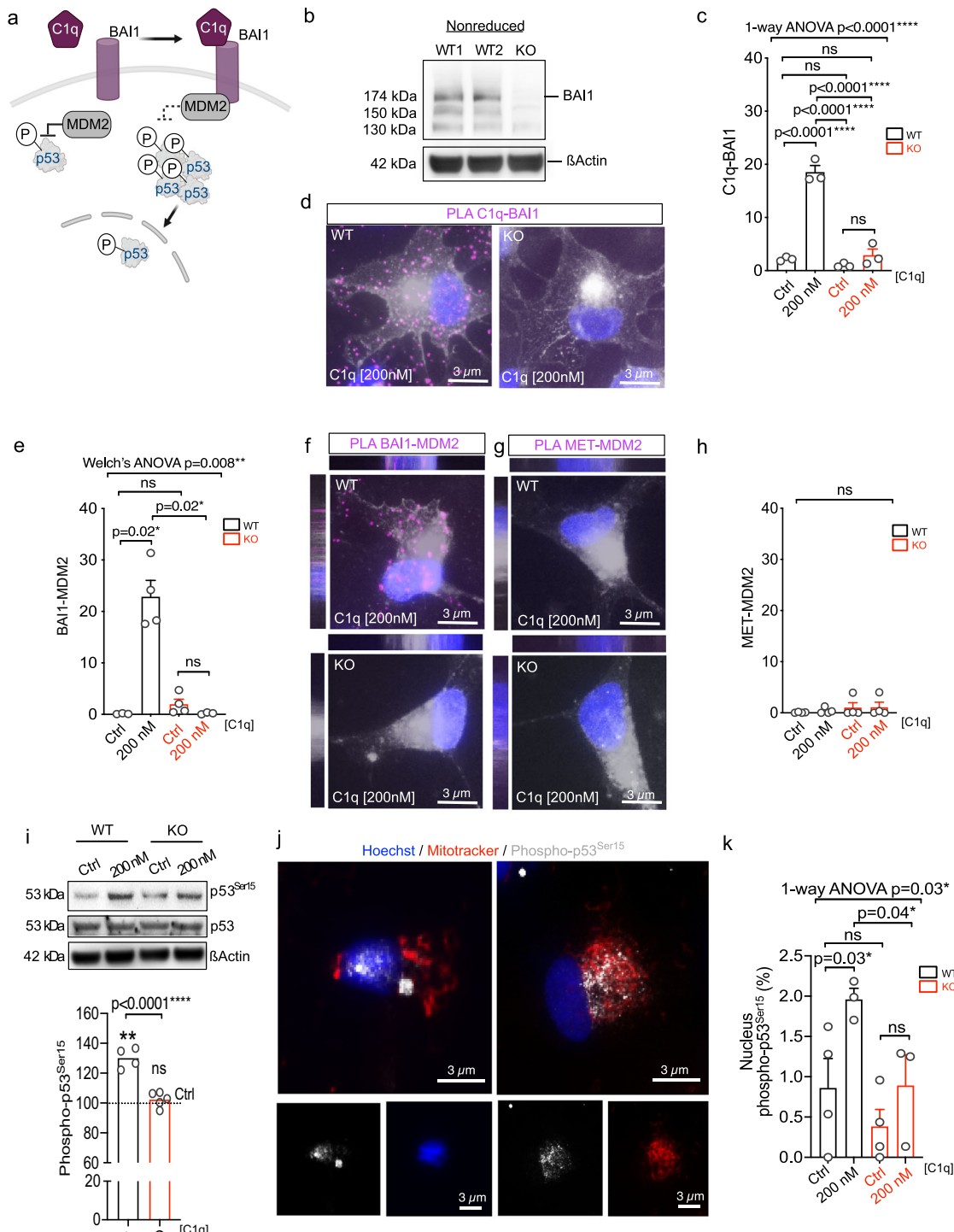

**Fig. 3 | C1q-driven p53 signal transduction is BAI1-dependent. a** Schematic of the tested hypothesis for BAI1-driven C1q mechanism of action in activation of p53 pathway in hNSC. Created in BioRender. Piltti, K. (2025) https://BioRender.com/rvswasg. **b** Loss of full-length 174 kDa BAI1 in BAI1 KO vs. WT hNSC visualized by Western blot ($n = 2$). **c–h** PLA analysis of protein interaction between C1q-BAI1 (**c**), BAI1-MDM2 (**e**), or MET-MDM2 negative control (**h**) in C1q-treated vs. control BAI1 WT or KO hNSC ($n = 3$ or 4 as indicated); representative images of PLA signal (magenta) for C1q-BAI1 (**d**), BAI1-MDM2 (**f**), or MET-MDM2 (**g**) in C1q-treated BAI1 WT vs. KO hNSC. WGA white, Hoechst blue. **i** Western blot analysis of p53 normalized phospho-p53$^{Ser15}$ in C1q-treated BAI1 WT or KO hNSC relative to each non-

treated control (ctrl, dashed line) ($n = 4$ or 5 as indicated). **j** Example images of phospho-p53$^{Ser15}$ immunostaining in Hoechst+ nucleus vs. cytoplasm with Mitotracker+ mitochondria. **k** Quantification of phospho-p53$^{Ser15}$ localization in nucleus in C1q-treated BAI1 WT or KO hNSC relative to each control ($n = 3$ or 4 as indicated). Mean ± s.e.m., **c**, **k** 1-way ANOVA with post hoc test, **e** Unequal variance Welch's ANOVA with post hoc test, **h** Kruskal-Wallis test, **i** 1-sample 2-tailed t-test and unpaired 1-tailed t-test. $n$ = biologically independent experiments. ns = not significant; *$p \leq 0.05$; **$p \leq 0.01$; ***$p \leq 0.001$; ****$p \leq 0.0001$. Exact p-values and $n$ available in Source Data. See also Supplementary Figs. 2 and 3.

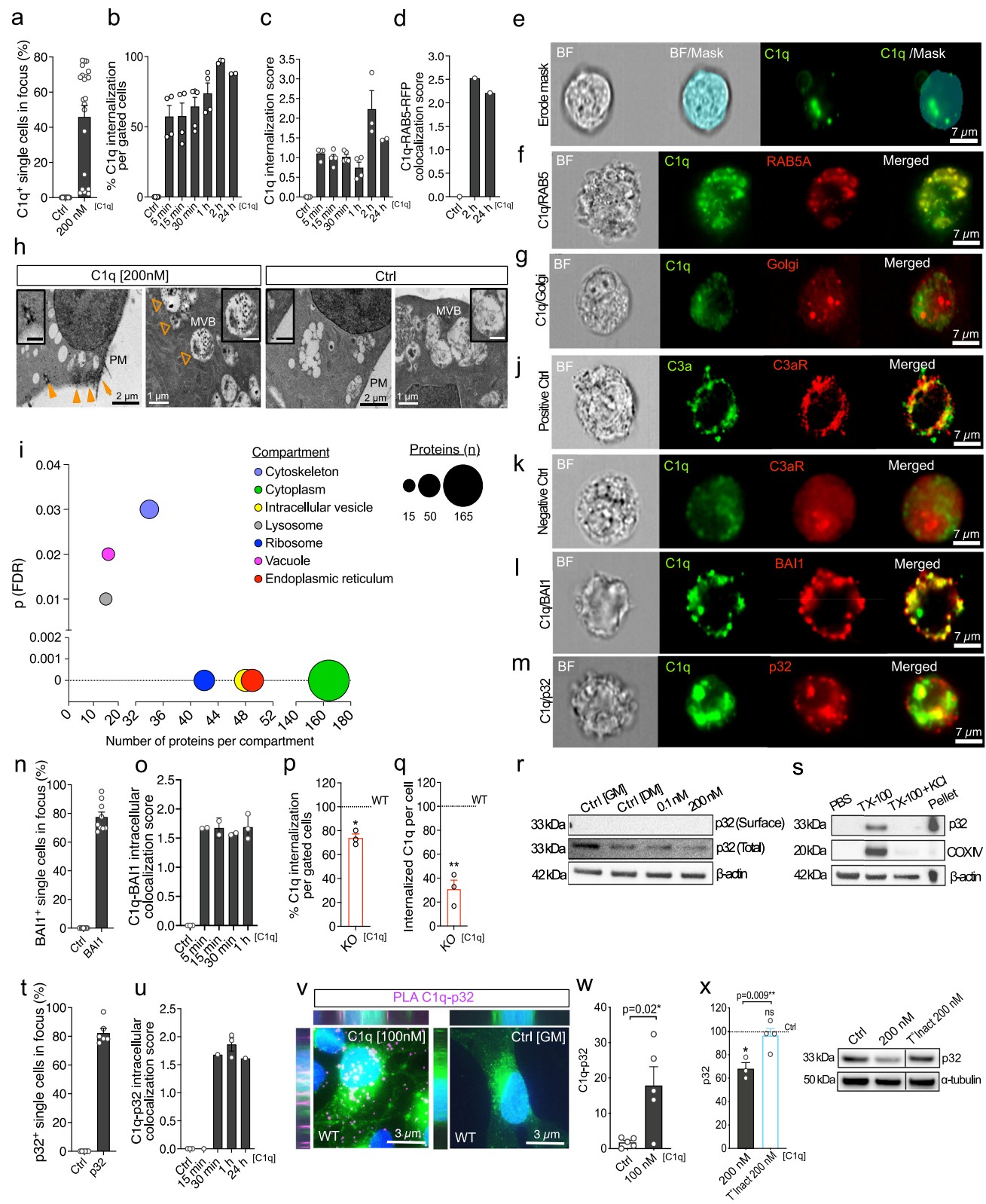

protein analysis (Fig. 5e), and mitochondrial p32 localization (Supplementary Fig. 5c–e) in BAI1 KO vs. WT cells. Critically, p32 transcription was unaltered by C1q treatment in WT cells, and p32 protein levels were unaltered in non-treated (baseline) WT vs. KO cells (Supplementary Fig. 5a, b). His-tag pull-down of purified human proteins in situ demonstrated a direct interaction between C1q-BAI1 and C1q-p32, as well as BAI1-p32, and that C1q-BAI1-p32 can be pulled down as a complex (Fig. 5f). Together, these data show that BAI1-mediates

endocytosis of C1q enabling intracellular protein interactions between C1q-p32, BAI1-p32, or C1q-BAI1-p32, decreasing total p32.

## C1q drives aerobic glycolysis via BAI1

p32 participates in several cellular functions, however, its most prominent role is to regulate mitochondrial metabolism. p32 cleavage or KO drives a shift from OXPHOS to glycolysis as well as loss of proliferation[56,57,59,60]. We therefore hypothesized that C1q-treated

**Fig. 4 | Endocytic internalization of C1q–BAI1 complex enables intracellular C1q–p32 interaction and increases p32 instability. a–c, e** Imagestream analysis of C1q internalization kinetics in C1q-treated vs. non-treated hNSC (**a–c**). Component masking strategy (**e**) (n = 2, 3, 4, or 5 as indicated; ≥3500 cells each). **d, f, g** Co-localization of C1q with Rab5a-RFP+ endosomes (**d, f**) vs. Golgi-RFP (**g**) (n = 1 per timepoint; ≥7000 cells each). **h** C1q-immunogold particles in endocytic structures at plasma membrane (PM; arrows) and late endosomal multivesicular bodies (MVB; arrowheads) selectively in C1q-treated hNSC (n = 2). Inset scale 500 nm. **i** Bubble plot showing subcellular enrichment of prey proteins in C1q-treated vs. non-treated hNSC in cellular compartments involved in receptor-mediated endocytic trafficking (x-axis, number of proteins, y-axis, significance; nanoLC-MS/MS, n = 4; FDR p ≤ 0.05). **j, k** Intracellular ligand-receptor co-localization in positive control C3a-C3aR (**j**) vs. mis-matched negative control C1q-C3aR (**k**) (n = 1 per timepoint; ≥7,000 cells each). **l, n, o** Time-course of C1q–BAI1 complex internalization in BAI1+ hNSC (n = 2 or 3 as indicated; ≥6900 cells each). **p, q** Proportion of cells with

intracellular C1q (**p**) and C1q amount within the cells (**q**) in C1q-treated KO hNSC relative to treated WT (dashed line; n = 3; ≥3000 cells each). **r, s** p32 in total protein vs. enriched cell surface fraction in hNSC (**r**), or in PBS-soluble, TX-100, TX-100+ KCl fractions vs. pellet (**s**), (n = 1). **m, t, u** C1q–p32 intracellular interaction kinetics (**u**) in C1q-treated hNSC (n = 1 or 3 per timepoint as indicated; ≥2000 cells each). **v, w** Representative C1q–p32 PLA images (**v**) and quantification (**w**) in C1q-treated vs. control hNSC (n = 5). PLA magenta, WGA green, Hoechst blue. **x** Western blot analysis of total p32 in hNSC treated with C1q or T°Inact C1q normalized to α-tubulin and controls (n = 3 or 4 as indicated); black line, non-adjacent lanes. Mean ± s.e.m., **i** one-sided Fisher's exact test, with Benjamini–Hochberg correction for multiple comparisons, **p, q** 1-sample 2-tailed t-test, **w** unpaired 1-tailed Welch's t-test, **x** unpaired 1-tailed t-test, and 1-sample 2-tailed t-test. n = biologically independent experiments. ns = not significant; **p ≤ 0.01; ****p ≤ 0.0001. Exact p-values, n, and full blots available in Source Data.

hNSC would exhibit a BAI1-dependent C1q-mediated glycolytic shift in cell metabolism (Fig. 5g) consistent with NSC quiescence[61–63]. We tested whether C1q treatment induces a glycolytic shift in hNSC, evaluating gene expression profile and NAD/NADPH ratio. RNAseq analysis followed by Gene Set Enrichment Analysis (GSEA) revealed a significant upregulation of glycolysis transcripts in C1q [200 nM] vs. control hNSC 48 h post-treatment (Fig. 5h, i). In parallel, timecourse analysis of NADH/NAD+ redox ratio using phasor two-photon fluorescence lifetime imaging microscopy (FLIM) identified a glycolytic shift after C1q [200 nM] treatment (Fig. 5j, k, Supplementary Fig. 6a)[64–66]. Timecourse analyses of mitochondrial membrane potential (Δψm) and reactive oxygen species (ROS) generation indicated that this outcome was not associated with mitochondrial dysfunction detectable as a loss of Δψm or broad increase in ROS (Supplementary Fig. 6b, c). Thus, C1q [200 nM] treatment shifts hNSC metabolism from OXPHOS to aerobic glycolysis.

We next investigated the effect of BAI1 KO on hNSC mitochondria fission, which is associated with both p32 and increased glucose metabolism[59,67], as well as on Seahorse metabolic analysis. Non-treated BAI1 WT and KO hNSC exhibited comparable baseline parameters (Supplementary Fig. 7a–c), however, a small shift in bioenergetics was detected in BAI1 KO hNSC (Supplementary Fig. 7c); data was therefore normalized to non-treated controls within each line. C1q [200 nM]-treated BAI1 WT hNSC exhibited increased mitochondrial fission (Fig. 5l, m), decreased oxygen consumption rate (OCR) (Fig. 5n), and increased extracellular acidification rate (ECAR) (Fig. 5o) vs. non-treated controls, each of which was reversed by BAI1 KO. Neither C1q treatment nor BAI1 KO affected hNSC ROS (Supplementary Fig. 7a), H+ proton leak, coupling efficiency, or non-mitochondrial oxygen consumption (Supplementary Data Fig. 7d), suggesting a specific effect on mitochondrial energy metabolism. These data provide evidence that C1q directly influences neural cell lineage bioenergetics, revealing a BAI1-dependent mechanism for these effects.

### BAI1 KO rescues C1q-induced proliferation decline in hNSC

In all, these data identify that C1q drives a decline in hNSC proliferation caused by reversible G0/G1 cell cycle arrest that is consistent with quiescence, and is mediated by at least two parallel mechanisms involving MDM2-p53 and p32. To confirm the relevance of BAI1 to cellular function we used proliferation and self-renewal as outcome measures. In vitro blockade of BAI1 using a characterized BAI1-neutralizing antibody (R&D systems, AF4969), as well as BAI1 KO, reversed C1q-mediated hNSC proliferation decline (Fig. 6a, b). Consistent with these results, BAI1 KO hNSC exhibited a dramatic reversal of C1q-driven loss of cell divisions that maintain the stem cell pool (Fig. 6c), suggesting restoration of a self-renewing mode of division.

We next sought to validate the biological significance of BAI1 in hNSC in vivo, selecting acute spinal cord injury (SCI) as the study paradigm. C1q increases after CNS trauma such as SCI[68,69]. We analyzed

local C1q expression by different cell types using published scRNAseq data[70] as well as total protein in our injury model. Nearly all immune cell populations and spinal cord cell types upregulated C1qa mRNA expression by 3–7 days post-injury (dpi; Fig. 7a, b), with C1q protein peaking in the SCI epicenter 3–24 hours post-injury (Fig. 7c), suggesting that both BSB disruption and local cell populations contribute to bioavailable C1q after SCI. We have also demonstrated that blockade of C1q at the lesion epicenter modulates the recruitment of transplanted hNSC after acute SCI[5,20], we therefore predicted that BAI1 KO would increase early hNSC engraftment. BAI1 WT and KO hNSC were transplanted into an acute SCI model, and proliferating hNSC were identified by co-labeling for STEM121 and BrdU at 2 weeks post-transplantation. Stereological analysis revealed a significantly greater number of BAI1 KO vs. WT hNSC in this paradigm (Fig. 7d–f). BAI1 KO did not modulate migration in vitro or in vivo (Supplementary Fig. 2j and Supplementary Fig. 7e, f). These data show that BAI1 KO rescues proliferation of transplanted hNSC, consistent with a reduction in quiescence signaling, and supporting a biologically significant role for C1q and BAI1 in the negative regulation of NSC in the CNS.

## Discussion

The traditional role of C1q in the immune system is as a recognition molecule for C1 complex and classical complement cascade activation; however, an emerging literature suggests that extracellular C1q can act as a single ligand independent of the complement cascade[71,72]. Five plasma membrane proteins have been recently identified as candidate receptors for C1q in hNSC[20] indicating that C1q may directly signal through these to modulate cellular functions. One of these is the GPCR BAI1. Here we show that C1q binds cell surface BAI1, and that the C1q-BAI1 complex is internalized and trafficked along the endocytic pathway, shifting NSC transcriptional state towards quiescence, and controlling two parallel downstream regulators of cell cycle, p53 and p32 (Fig. 7g). This mechanism is C1, C1s, and complement cascade-independent. These data establish direct mechanistic links between C1q and p53 regulation of proliferation, as well as between C1q and p32 regulation of mitochondrial function and glycolysis, consistent with quiescence[61–63]. In sum, we identify that C1q interaction with BAI1 drives proliferation decline and metabolic reprogramming in NSC and establish a role for C1q in stem cell quiescence.

A potential role of extracellular C1q in cell proliferation and function has also been investigated in muscle satellite stem cells[15] and CD8 + T cells[55,73,74]. Critically, the mechanisms we identify here in CNS cells are distinct, highlighting the importance of our findings and that different tissues employ C1q signals for common purpose but via different mechanisms. In muscle, extracellular C1q drives proliferation decline in satellite stem cells via a mechanism dependent on C1 complex and proteolytic activation of C1s to mediate downstream Wnt signaling[15]. In contrast, we show that C1q interaction with BAI1 and downstream modulation of p32/p53 signaling are required in hNSC. In

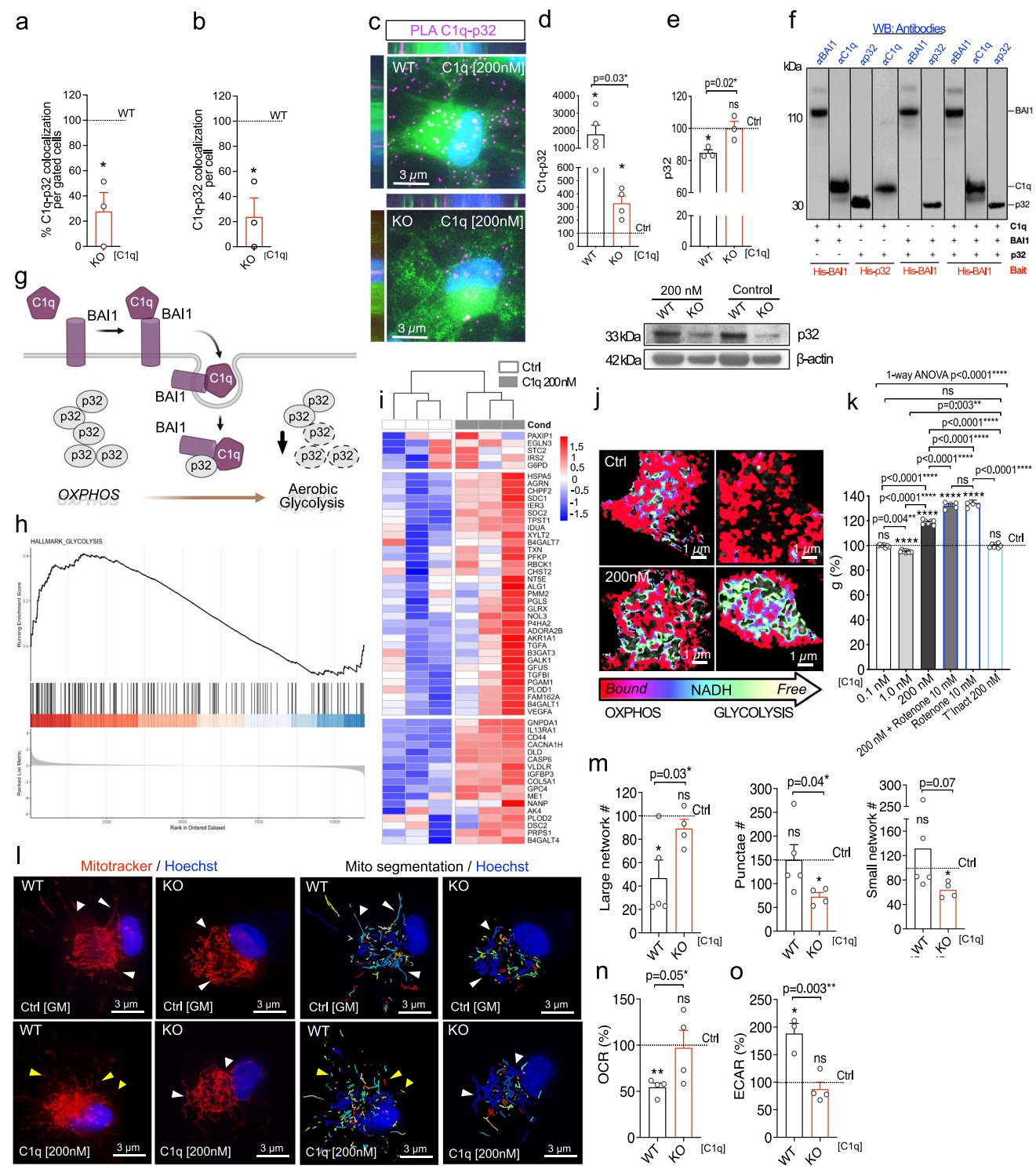

the immune system, extracellular C1q independent from the complement cascade decreases CD8 + T cell proliferation, which restrains autoimmunity[55]; these effects are driven by C1q interaction with cell surface p32. In contrast, we show that p32 in NSC is localized intracellularly, and that C1q interaction with p32 requires internalization of C1q-BAI1 complex. Despite the difference in location where C1q initially engages with p32, C1q-p32 interaction leads to metabolic reprogramming and a shift in cellular activation state in both CD8 + T cells and NSC. Whether C1q interaction with or modulation of p53 also plays a role in T cells has not been reported. Similarly, whether p32 alone is responsible for C1q endocytosis in T cells is unknown. Finally, it

remains to be elucidated whether the mechanisms identified here intersect with the pathways established in other tissues.

The effects of C1q on each of these tissues reflect increases in extracellular C1q levels associated with aging, injury, and disease[15,55,75]. Importantly, both local cells and anatomical location/extracellular matrix composition may modulate C1q bioavailability. In the CNS, microglia are the principal source of C1q in the normal and aging brain[10], secreting homeostatic concentrations of C1q of ~1 nM. We identify that 1 nM C1q induces a significant decrease in free NAD/NADH ratio and shift towards OXPHOS in NSC; this result is consistent with the role of oxidative metabolism in NSC activation and

**Fig. 5 | C1q-p32 intracellular interaction drives p32 instability and aerobic glycolysis in a BAI1-dependent manner. a, b** Imagestream analysis of intracellularly co-localized C1q–p32; Cell% (**a**) and the amount (**b**) in C1q-treated KO relative to treated WT hNSC (dashed line; *n* = 3; ≥2000 cells each). **c, d** Representative C1q–p32 PLA images in C1q-treated WT vs. KO hNSC (**c**) and quantification (**d**) relative to each control (*n* = 4 or 5 as indicated). PLA magenta, WGA green, Hoechst blue. **e** Western blot of total p32 in C1q-treated WT and KO hNSC normalized to β-actin and each control (*n* = 3). **f** In situ His-tag pull-down showing direct interaction between C1q-BAI1, C1q-p32, and BAI1-p32 (*n* = 2). 4 pull-downs in 3 parallel blots; black line, non-adjacent lanes or separate blots. **g** Schematic of hypothesized metabolic outcome of BAI1-dependent C1q-p32 interaction leading to reduced p32 stability. Created in BioRender. Piltti, K. (2025) https://BioRender.com/rvswasg. **h, i** GSEA plot enrichment of HALLMARK_GLYCOLYSIS pathway (**h**, FDR p ≤ 0.05) and heat map of contributing transcripts (**i**) after C1q treatment (RNAseq, *n* = 3).

Red up-regulation, blue down-regulation. **j, k** Representative FLIM images of bound vs. free NADH/NAD+ in C1q-treated and control hNSC (**j**) and redox ratio in C1q vs. Rotenone-treated cells relative to controls (**k**) (*n* = 5, 7 or 9 cells as indicated). **l, m** Representative images of mitochondria morphology in non-treated and C1q-treated Mitotracker+ WT or KO hNSC (**l**) and the analysis shown relative to controls (**m**) (*n* = 4 or 5 as indicated). **n, o** OCR (**n**) and ECAR (**o**) in C1q-treated WT or KO hNSC relative to controls (*n* = 3 or 4 as indicated). Mean ± s.e.m. **a, b** 1-sample 2-tailed t-test, **d, n** unpaired 1-tailed Welch's t-test with 1-sample 2-tailed t-test, **e, o** unpaired 1-tailed t-test and 1-sample 2-tailed t-test, **h** two-sided permutation-based enrichment score test with Benjamini–Hochberg correction for multiple comparisons, **m** unpaired 1-tailed t-test or unpaired 1-tailed Welch's t-test with 1-sample 2-tailed t-test, **k** 1-way ANOVA with post hoc and 1-sample 2-tailed t-test. *n* = biologically independent experiments. ns = not significant; *p ≤ 0.05; **p ≤ 0.01; ***p ≤ 0.001; ****p ≤ 0.0001. Exact p-values, *n*, and full blots available in Source Data.

differentiation[16,17], and with our previous reports that 1 nM C1q increases NSC oligodendroglial differentiation[5,20]. In contrast, BBB/BSB breakdown in association with trauma or disease can also expose the CNS to high C1q levels derived from blood plasma and/or increased local synthesis. We report that the threshold for C1q to induce the observed effects is between 25 and 100 nM. While much of the C1q in plasma is present in C1 complex, an estimated 20% is present as free C1q[76,77]. With plasma C1q values reported in the literature spanning 122 nM–600 nM[78], corresponding to 24–119 nM free C1q, within the range tested in this study. Thus, loss of BBB/BSB integrity may be sufficient to disrupt homeostasis in the neurovascular NSC niche. Together, these data suggest that C1q in the CNS microenvironment regulates NSC metabolism and the balance between NSC quiescence and activation/differentiation[79–81], contributing to stem cell homeostasis, impacting pathological processes, and modulating repair capacity. Overall, these findings align with recent studies suggesting that a shift to quiescence in aging endogenous NSC is driven by niche-derived inflammatory signals[82].

Importantly, we have previously shown that GPCR inhibition via pertussis toxin (PTX) blocked C1q-induced proliferation decline but not C1q-induced migration, whereas Erk inhibition via PD98059 modulated both migration and proliferation[20]. These data suggest that cell surface GPCR/s such as BAI1 play a key role in C1q proliferation signaling, and also that the Erk1/2 pathway is involved in multiple cellular functions in hNSC[83–85]. In addition, BAI1 may have other ligands that can also trigger quiescence in hNSC. Given that C1q internalization was only partially inhibited in BAI1 KO, other candidate C1q receptors could play a role in C1q internalization and modulation of cell proliferation[83–85]. In addition, our data demonstrate a direct interaction not only between C1q-BAI1 and C1q-p32 but also BAI1-p32, enabling pulldown of C1q-BAI1-p32 as a complex. The role of BAI1-p32 interaction as well as how dynamic C1q-BAI1-p32 complex might be within the cell remains uncertain. It is plausible that C1q-BAI1 signaling also modulates other aspects of brain function, particularly in neurological disease and injury. For example, p53, BAI1, and C1q have all been linked to malignant brain tumors[86,87]. C1q-p32-driven metabolic changes may also have implications for cancer, as increases in C1q and glycolysis are associated with malignant tumors[87,88]. Finally, both C1q and BAI1 have also previously been identified roles in remodeling dendrites, synaptogenesis, and synaptic plasticity[38,89–93]. Given that endogenous NSC as well as CNS lineage cells express the BAI1-receptor, a broader role for these pathways is likely, but remains to be investigated. Thus, these findings are relevant for a multiplicity of CNS conditions involving BBB/BSB dysfunction or local accumulation of C1q.

## Methods

### Stem cell lines

Two multipotent human male fetal CNS-derived stem cell lines (hNSC; Stem Cells Inc. 2491.2[94] and UCI161, UC Irvine, Anderson AJ) were isolated from gestational week 16–20 fetal brain using comparable derivation methods and enriched for CD133 stem cell marker[94,95]. BAI1 WT and KO hNSC were generated using CRISPR-Cas9 editing of UCI191 hNSC (Supplementary Fig. 2). All hNSC lines displayed neurosphere formation, chemotactic responsiveness, and stable multipotency for >20 passages, generating neurons, oligodendrocytes, and astrocytes[94,95]. A multipotent mouse NSC line was derived from single Fucci mouse[21] cortices at E11 (Fucci-mNSC) as previously described in ref. 23. Both human and mouse NSC showed reduced proliferation in response to C1q. Sex was not considered in all elements of in vitro analysis of human cells because of the volume of data collected. While sex-based differences between NSC lines were not performed in this study, both male and female hNSC lines express BAI1 and responded to C1q (Supplementary Fig. 7g, h).

### Mice

Ten-week-old immunodeficient female Rag-1-mice (*n* = 14; B6.12957-Rag1tm1Mom/J; Jackson Laboratory, 002216) were used as the SCI model. Sex was not considered in the study design; Females were selected to minimize bladder complications and urolithiasis common in male mice following SCI[96,97]. Mice were housed in cages with Alpha-dri bedding, nestlets, and housing under IACUC approved conditions guidelines with a 12 h dark/light cycle and access to food and water ad libitum. Fucci reporter mice[23,98] were bread to generate Fucci-mNSC line for live imaging of cell cycle dynamics.

### Proliferation, neurosphere formation, and RealTime-Glo Annexin V multiplex assay

Human and mouse NSC were cultured as free-floating neurospheres in growth medium (GM) as previously described in ref. 94. BrdU (AmershamTM Cell Proliferation Kit, GE Healthcare, 45-000-837) and EdU proliferation assays (Invitrogen, C10338) were performed according to the manufacturer instructions in monolayer cultures on poly-L-ornithine (PLO) and laminin-coated chamber slides without (non-treated controls) or with purified human C1q (MyBioSource, MBS147305) or purified human C1 (CompTech, A098). BrdU or EdU was incorporated for 48 h before fixation (4% PFA) and immunostaining. Hoechst (1 μg/mL; Invitrogen, H3570) was used as a nuclear counterstain. For preconditioning, hNSC were exposed to 200 nM C1q for 48 hours (h), followed by washout and 48 h BrdU incorporation. Biologically independent experiments were performed with technical duplicate wells. Images were acquired using Olympus IX, Zeiss LSM900 or ApoTome, or Keyence BZ-X microscopes with 20x objective and analyzed using Imaris (Andor Technology, version 7.5.2. or 9.1.2) by investigators blinded to experimental conditions/groups. BrdU/EdU/Hoechst% per each captured image/independent experiment were tested for outliers using ROUT (Q = 10%) in Prism (GraphPad, version 10); outliers are listed in the Source Data.

For clonal neurosphere assays, free-floating neurospheres were dissociated into single cells (Supplementary Methods) and plated into a 96-well plate (15 cells/well) in GM, with or without C1q. Sphere-

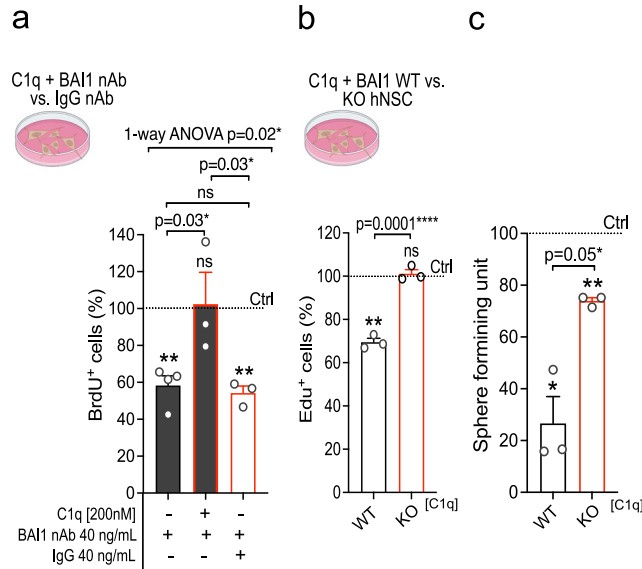

**Fig. 6 | BAI1 KO rescues hNSC proliferation decline and self-renewal during C1q treatment in vitro. a** BrdU analysis of C1q and BAI1 nAb vs. IgG nAb co-treatment effect on hNSC proliferation relative to non-treated controls (dashed line; *n* = 3 or 4 as indicated)#. **b** EdU analysis in C1q-treated BAI1 WT or KO hNSC relative to each control (*n* = 3)#. **c** Clonal neurosphere assay of self-renewal capacity in C1q-treated BAI1 WT or KO hNSC relative to each control (*n* = 3). Mean ± s.e.m., **a** 1-way ANOVA with post hoc test and 1-sample 2-tailed t-test, **b** unpaired 1-tailed t-test and 1-sample 2-tailed t-test, **c** Mann Whitney1-tailed test and 1-sample 2-tailed t-test. *n* = biologically independent experiments. ns = not significant, *p* > 0.05; *$p \le 0.05$; **$p \le 0.01$; ****$p \le 0.0001$. Exact p-values and *n* are available in Source Data. #Illustration created in BioRender. Piltti, K. (2025) https://BioRender.com/rvswasg.

forming units were calculated at 2.5 weeks (w) post-plating as the proportion of new neurospheres (⌀>50 μm) to plated cells[99], and neurosphere sizes were measured using ImageJ (NIH). For C1q priming assays, free-floating neurospheres were co-cultured with or without C1q 48 h or for up to 6–8w followed by wash out and plating as a monolayer for BrdU or a low adherence for neurosphere initiation in GM. Post-C1q wash, BrdU was incorporated for 48 h, and sphere forming units were analyzed 5w later.

RealTime-Glo Annexin V (Promega, JA1011) apoptosis and necrosis assays were performed according to the manufacturer instructions with or without C1q. hNSC treated with 5 μM ionomycin (Sigma-Aldrich, I3909) or 0.8 μM stauroporine (Sigma-Aldrich, S6942) were used as positive controls. Biologically independent experiments were performed with technical triplicates. Quantification of all experiments were performed by investigators blinded to experimental conditions/groups. More details available in the Supplementary Methods.

## C1q-, C1s- and BAI1 neutralizing antibodies and C1 inhibitor
hNSC were plated as monolayers on PLO/laminin-coated chamber slides with or without C1q and co-cultured with C1q nAb goat anti-human C1q (Quidel, A031) or normal goat IgG control (R&D systems, AB-108-C), complement component C1s antibody (Novus Biologicals, NBP1-52122) or mouse IgG control (Invitrogen, 1903), C1 inhibitor (Millipore, GF178), human BAI1 Antibody (R&D systems, AF4969) or normal sheep IgG control (R&D systems, 5-001-A) for 48 h in presence of BrdU or Edu. PFA 4%-fixed, immunostained, and imaged samples were analyzed using Imaris software by investigators blinded to experimental conditions/groups. Biologically independent experiments were performed with technical duplicate wells. BrdU/EdU/Hoechst% per each captured image/independent experiment were

tested for outliers using ROUT (Q = 10%) in Prism; outliers are listed in the Source Data.

## Fluorescence live cell imaging and tracking
Time-lapse image acquisition of Fucci-mNSC monolayers cultured in custom-made PDMS without (non-treated control) or with 0.1 nM or 200 nM C1q and analysis were performed as previously described[23]. Each biologically independent experiment included 5 technical replicate microwells per group, one microwell per group with comparable cell confluences was captured and analyzed per each independent experiment. The total number of cells and proportions of G1 or S, G2, or M cell phases were quantified at 0 h, 48 h, 72 h, 96 h, and 7 days (d).

For single-cell tracking of NSC survival, 30 Fucci-mNSC in G1 phase were randomly selected per biologically independent experiment and tracked up to 7 days using Imaris software as previously described in ref. 23. Cell death was calculated as the average percentage of dead cells out of the total tracked cells (example in Supplementary Fig. 1d, e). A total of 4 biologically independent experiments, 120 cells/group, were analyzed by investigators blinded to experimental conditions/groups.

## PathScan intracellular signaling array and Western blotting
Intracellular signaling arrays (PathScan, Cell Signaling Technology, 7744) were analyzed from previously published data[20]. The data shown are from hNSC treated with C1q at 0.1 nM, 1.0 nM, or 200 nM concentrations for 60 min, normalized to internal positive controls and non-treated hNSC controls.

Western blot analyses were performed using protein lysates collected from non-treated, C1q or temperature-inactivated C1q-treated hNSC on 6-well plates. Band densities were quantified using ImageJ (NIH) and normalized to loading controls β-actin, α-tubulin, or total p53 (phospho-p53$^{Ser15}$) and non-treated controls. Each biologically independent experiment consists of one well. Primary antibodies and dilutions used were as follows: Tubulin β3 (1:500, Biolegend, TUBB3, 801202), Glial Fibrillary Acidic Protein (1:5000, Dako, Z 0334AB_10013382), Glial Fibrillary Acidic Protein δ (1:500, Millipore, AB9598), S100β (1:200, Abcam, Ab52642), vimentin (1:1000, Abcam, Ab24525), nestin (1:1000, Abcam, Ab11306), SOX2 (1:200, R&D systems, AF2018), Phospho-p53 (Ser15) (1:300, Cell Signaling, 16G8, 9286), p53 (1:500, Cell Signaling, 9282 s), GC1qR (p32) (1:100, Abcam, Ab24733), MTCO1 COX IV (1:1000, Invitrogen, 459600), BAI1 (1:1000, R&D systems, AF4969), BAI1 (1:1000, Novus Biologicals, NB110-81586), BAI1 polyclonal antibody (1:1000, Invitrogen, PA5-102069), β-actin (1:2500, Sigma-Aldrich, A1978), or α-tubulin (1:7000, Abcam, DM1A, Ab7291). For pulldown analysis, we used antibodies against BAI1 (1:1000, R&D systems, AF4969), C1q (1:500, Abcam, Ab71940), and p32 (1:2000, Abcam, Ab24733). Secondary antibodies used were horse-radish peroxidase (HRP)-conjugated secondary antibodies at 1:5000 dilution (Fisher Scientific, NA931, NA934, or NA935) or (ThermoFisher, 31402, A16041, or A16011). For quantitative comparisons, experimental control and sample lanes, as well as loading controls for each biologically independent experiment, were analyzed from the same membranes. Samples with technical errors were excluded from the study. Normalized phospho-p53Ser15 and p32 bands/independent experiment were tested for outliers using Grubb's outlier test (Alpha 0.2) in Prism. Exclusions, outliers, and unprocessed scans of the most important blots with loading controls are provided in the Source Data.

## RNAseq and single-cell RNAseq
hNSC were plated at 500,000 cells/well as a monolayer in 6-well plates and cultured with or without 200 nM C1q in differentiation medium (DM) for 48 h. Each biologically independent experiment consists of one well. RNA was extracted and quantified for quality (Supplementary Methods). Library construction and sequencing were carried out at the UCI GHTF using a TruSeq RNA Library Prep Kit and Illumina HiSeq

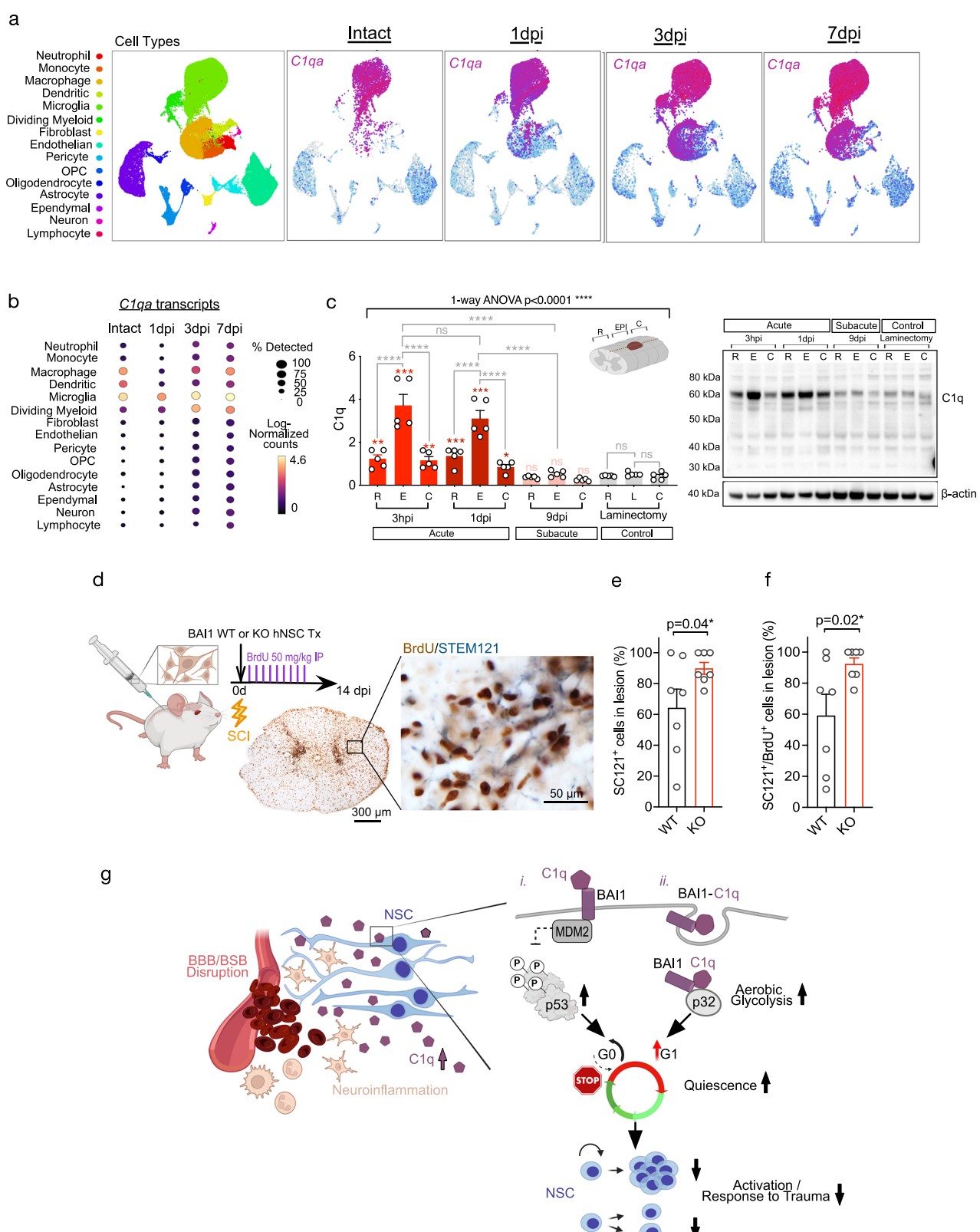

4000 (Illumina), generating ~30 million paired-end reads. Before read mapping, each sample was quality-checked with FastQC[100]. Ribosomal contamination was removed using the SortMeRNA[101], and low-quality bases (Q < 30) were filtered out with Trimmomatic tool[102]. RNAseq reads were mapped to the human genome GRCh38 version 105 using the STAR aligner, uniquely mapped reads were summarized using featureCounts[103] and differential gene expression analysis was conducted on TMM normalized counts using EdgeR[104]. Significant genes were identified with an FDR-adjusted $p \leq 0.05$. Gene Set Enrichment Analysis (GSEA version 4.3.2)[105] was performed using the hallmark gene set collection from Molecular Signature Database (MSigDB)[106]. Pathway enrichment analysis of quiescence, activation, and neural differentiation-associated transcripts was analyzed using GSEA for each manually curated gene set. Normalized enrichment

**Fig. 7 | BAI1 KO rescues hNSC proliferation decline and engraftment after transplantation into acute CNS injury with elevated C1q. a** UMAP visualization of major cell types expressing *C1qa* in intact and injured spinal cords, highlighting potential injury-associated changes in *C1qa* expression[70]. **b** Dot plot showing *C1qa* transcript abundance and temporal dynamics across cell types; dot size indicates transcript percentage, color intensity log-normalized expression[70]. **c** Western blot analysis of β-actin normalized total C1q protein in laminectomy controls and injured spinal cords at 3 h, 1 d, and 9 d post-contusion[20]. Lysates were collected from rostral (R), epicenter (E), and caudal (C) sections (*n* = 5 mice per timepoint). **d** Experimental paradigm for testing biological significance of BAI1 in acute spinal cord injury microenvironment associated with blood plasma-derived C1q increase due to injury-associated BBB/BSB breakdown[#]. **e, f** Unbiased stereological analysis of STEM121+ human cell engraftment (**e**) and proliferation (**f**) in spinal cord sections aligning with injury lesion after in vivo transplantation of BAI1 WT vs. KO hNSC (*n* = 7 mice per group). **g** Proposed model of C1q–BAI1 signaling in NSC regulation following CNS injury[#]. Mean ± s.e.m., **c** 1-way ANOVA and post hoc test, **e** unpaired 1-tailed Welch's t-test, (**f**) 1-tailed Mann Whitney test. ns = not significant; *$p \le 0.05$; **$p \le 0.01$; ***$p \le 0.001$; ****$p \le 0.0001$. Exact *p*-values are available in Source Data. [#]Illustration created in BioRender. Piltti, K. (2025) https://BioRender.com/vgqd51n.

---

scores (NES) were visualized using the ggradar R package. Gene sets between the groups were considered statistically significant at a multiple comparison p-value of 0.05 and visualized using heatmaps. All analyses were performed by investigators blinded to experimental conditions/groups. Local C1q expression by different spinal cord cell types in intact mice vs. 1, 3, or 7 d post-SCI was analyzed using previously published scRNAseq data[70].

### CRISPR Cas9 hNSC BAI1 genetic deletion

Tissue-derived multipotent hNSC are a heterogenous stem cell population that in long-term culture with an increasing number of passages exhibit an increased risk for undesirable profile changes to a gliogenic phenotype. Therefore, isogenic clone selection carries a heightened risk of inadvertent deletion of cell subpopulations or types. To address this, we developed a two-stage genetic modification strategy using CRISPR cas9 and FACS to create BAI1 WT and KO hNSC. Based on our initial analysis, 80% of unmodified hNSC were BAI1$^+$, and 20% were BAI1-.

Using a custom BAI1 donor template (Supplementary Fig. 2a) and a BAI1 CRISPR plasmid (Applied Stem Cells, cat no. P0049-P001), we introduced premature stop codons into first open reading frame of BAI1 within exon 2 - along with a floxed CMV-mScarlet-P2A-PuroR cassette leading to truncated BAI1 at amino acid 52, reducing the size of BAI1 by 96.7% to effectively create precise genetic KO in hNSC (Supplementary Fig. 2b). After positive selection of mScarlet-expressing KO hNSC using FACS, CRE recombinase was added to BAI1 KO hNSC to remove the selection cassette (Supplementary Fig. 2b), causing the BAI1 KO hNSC to become mScarlet-negative. The cas9-expressing-BAI1 Guide RNA plasmid (gRNA sequence in Supplementary Fig. 2c) was custom-generated and validated in human HEK293 cells by Applied Stem Cells (cat no. P0049-P001). The BAI1 donor plasmid (Supplementary Fig. 2a) was constructed in-house (Supplementary Methods). Homozygous insertion of the frame-shift premature stop codons in BAI1 KO hNSC was verified via PCR of genomic DNA and Sanger sequencing (Supplementary Fig. 2d, e). Stable loss of total BAI1 protein (174 kDa) expression was confirmed under non-reducing conditions by Western blotting (Fig. 3b), and functional level, by performing PLA specific for the protein-protein interaction of BAI1 with its ligand, C1q, in both BAI1 KO and WT cells (Fig. 3c, d). Two different BAI1 KO lines were generated with the same method. No substantial differences between the cell lines generated were detected regarding their CD133$^+$ content, multipotency, migration, or proliferative capacity. BAI1 KO hNSC characterization is detailed in the Supplementary Methods.

### Proximity ligation assays

hNSC were plated on PLO-laminin or biolaminin (BioLamina, CT521-0501) coated 8-well glass chamber slides and treated with 100 nM or 200 nM C1q or equal volume of GM (non-treated control) for 30 min or 1 h as previously described in ref. 20. All biologically independent experiments were performed with technical duplicates. PFA 4%-fixed and wheat germ agglutinin (WGA)-conjugate 488 (Invitrogen) stained cells were permeabilized and blocked with 0.01% Triton X (Sigma Aldrich) and 2% Donkey Serum (Jackson Immunoresearch). PLA was performed using Duolink® In Situ PLA® Probes (Supplementary Methods) following the manufacturer instructions. PLA signal, presented as total punctae per cell with a positive signal, was quantified using Imaris from z-stacks of optical slices captured at 0.3 μm intervals using a 60x objective. Primary antibodies used were anti-C1q (1:100, Abcam, Ab71940), anti-BAI1 (1:500, Abcam, Ab135907), anti-MDM2 (1:100, Santa cruz biotechnology, SMP14, Sc-965), recombinant anti-Met (c-Met) (1:500, Abcam, Ab51067), or gC1qR rabbit (1:500, Abcam, ab270033). All analyses were performed by investigators blinded to experimental conditions/ groups.

### Immunocytochemistry and flow cytometry

hNSC monolayers fixed with 4% PFA were permeabilized and immunostained with primary antibodies against cleaved caspase-3 (1:100, Cell Signaling, Asp175, D3E9, Cat. No. 9602 s), phospho-p53 (Ser15) (1:400, Cell Signaling 16G8, 9286 s), recombinant anti-GC1qR (p32) (1:200, Abcam, Ab24733), or TOM20 (1:200, Cell Signaling, D8T4N, 42406S). Secondary antibodies conjugated with 555, 488, or 647 fluorochromes were used at 1:2000, 1:1000, or 1:500 dilution, respectively (Invitrogen, A131570, A312572, A21202, A21206, A31571, A31573, or A21447), and Hoechst 33342 (1 μg/mL; Invitrogen, H3570) was used as a nuclear counterstain. All biologically independent experiments were performed with technical duplicates. Images captured using a LSM900 (Zeiss) or BZ-X (Keyence) microscope with a 20x objective were analyzed with Imaris by investigators blinded to experimental conditions/groups. For phospho-p53Ser15 quantification, the cells with only nuclear label were considered positive. Samples with technical issues during the staining, such as drying out of the wells or wells with insufficient amount of antibody, were excluded from the analysis. Exclusions and outliers are listed in the Source Data.

For imaging flow cytometry, hNSC were plated on PLO-laminin-coated flasks and treated with 200 nM C1q or an equal volume of GM (non-treated control) for 5 min, 15 min, 30 min, 1 h, 2 h, or 24 h. For testing C1q colocalization in endosomes or Golgi, hNSC were transfected with either CellLight™ Early Endosomes-RFP (Invitrogen, C10587), or Golgi-RFP (Invitrogen, C10593). The next day, cells were washed and treated with 200 nM C1q for 24 h. Details for matched positive and mismatched negative ligand-receptor controls (Fig. 4j, k and Supplementary Fig. 4b–j) are in the Supplementary Methods. After treatment, cells were washed and detached using Trypsin/EDTA or non-enzymatic cell dissociation solution (Sigma-Aldrich, C5914). After treatment, the cells were aliquoted at a density of 20,000 cells/μL per tube and fixed using 2% PFA for 15 min on ice, followed by wash with 1% fish gelatin in 0.1 M TBS, blocking, and 1 min permeabilization using DPBS supplemented with 0.1% or 0.02% Triton X-100, 5% goat or donkey serum. Primary antibodies diluted in 0.1 M TBS wash buffer with 1% fish gelatin and 5% serum were incubated with cells overnight at +4 °C. Primary antibodies used were anti-C1q (1:50, Abcam, Ab71940), anti-human BAI1 (1:50, R&D systems, AF4969), anti-BAI1 (1:50, Lifespan Biosciences, LS-C120632), anti-human complement C3a (1:50, Millipore/Chemicon, CBL191), anti-C3aR (1:50, Abcam, Ab103629), anti-EGFR (Abcam, EP38y, ab52894), recombinant anti-GC1qR (1:50, Abcam, Ab24733). Secondary antibodies used (1:500) were conjugated with 555, 488, or 647 fluorochromes (Invitrogen,

A131570, A312572, A21202, A21206, A21141, A31571, A31573, A21240, or A21447). The BAI1 antibodies used for Imagestream, Western blotting, or PLA, all recognize the full-length BAI1 when evaluated using BAI1 overexpression lysate and empty vector negative HEK293T control lysate (Novus Biologicals; Supplementary Fig. 3a). Single cells were imaged using Amnis Imagestream x Mark II Imaging Flow Cytometer and analyzed using IDEAS version 6.2 (Cytek Biosciences)[51,107] and component masking (Fig. 4e) according to the manufacturer instructions. Analysis details are available in the Supplementary Methods. Gating (Supplementary Fig. 3b,c) were set using unstained negative control cells, secondary antibody only controls, single-labeled cells, and antibody compensation beads. For protein internalization, score ≥0.5 was considered internalized. Mean bright detail similarity R3 score ≥1.5 was considered colocalized. Spot count feature (Supplementary Fig. 4o) was used for measuring the quantity of internalized protein per cell. Time-course data was collected from one to four biologically independent experiments with one sample per each. Each biologically independent experiment consists of one flask.

### Immunogold electron microscopy

For electron microscopy, hNSC were cultured on 8-well chamber slides with or without C1q for 24 h. After washing with 0.1 M phosphate buffer, cells were fixed with a mixture of 0.25% glutaraldehyde (Ted Pella, Inc., #18420) in 2% PFA (Polysciences, #00380) in 0.1 M PB with 50 mM sucrose and 0.4 mM $CaCl_2$ on ice followed by washes, blocking, and quenching with 1% NaBH4 in 0.01 M PBS. Cells were permeabilized with 0.2% Triton-X 100 and incubated with anti-C1q antibody (1:50; Abcam, Ab71940) overnight at 4 °C. The next day, nonspecific binding sites were blocked with wash buffer and incubated with 1.4 nm Nanogold-anti mouse Fab' fragment (1:50, Nanoprobes Inc., #2022) at room temperature for 2 h followed by fixation with 1% glutaraldehyde. Immunogold deposits were enhanced by incubation with ionic silver in citrate buffer (HQ Silver Enhancer, Nanoprobes Inc., #2012) followed by washes, post-fixing with 2% $OsO_4$, dehydration, and embedding in Epon. Sections cut at 80 nm thickness were counterstained with 4% uranyl acetate and lead citrate and examined with a JEOL 100CX transmission electron microscope (JEOL USA). Negatives were scanned using an AGFA Duoscan T1200, and images were processed using Photoshop (Adobe Systems). Data represents two biologically independent experiments with technical duplicates. All analyses were performed by investigators blinded to experimental conditions/groups.

### C1q biotinylation, cross-linking, co-immunoprecipitation, and LC-MS/MS

C1q was biotinylated using cell membrane-permeable EZ-Link NHS-SS-Biotin (ThermoScientific, PI21441). Biotinylated C1q (200 nM) or growth medium (control) was applied to cells for 15 min or 30 min at RT. Analysis was performed as four biologically independent experiments with one sample well per each. Intracellular crosslinking was performed with DSP (ThermoScientific, PI22586), followed by lysis in Pierce IP buffer (ThermoScientific, PI87787) and pull-down on Streptavidin Magnetic Beads (ThermoScientific, PI188816). Step-by-step protocols describing the procedures used in this study for have been deposited on protocols.io and can be accessed as follows: C1q biotinylation, cross-linking, and co-immunoprecipitation/pull-down dx.doi.org/10.17504/protocols.io.14egnrydpl5d/v1, denaturing on-bead-digestion dx.doi.org/10.17504/protocols.io.bp2l6zp91gqe/v1, and desalting dx.doi.org/10.17504/protocols.io.rm7vz92bxgx1/v1. Peptide digests were analyzed using an UltiMate 3000 RSLC system coupled in-line to an Orbitrap Fusion Lumos mass spectrometer. Separation was performed on a 50 cm Acclaim® PepMap RSLC column with a 4% to 22% gradient over 70 min at 300 nL/min. Mass spectra were acquired in a cycle of full Fourier transform scans (375–1500 m/z, resolution 120,000) followed by data-dependent MS/MS scans with HCD at 30% NCE. Ions selected

for MS/MS were dynamically excluded for 30 s. Protein identification and label-free quantitation were performed using MaxQuant version 1.6.0.16109. Further details available in the Supplementary Methods. Peptide spectra match and protein FDR were both set to 0.01. Data analysis utilized Differential Enrichment Pipeline (Bioconductor, DEP version 1.22)110. Samples treated with biotinylated C1q per each biologically independent experiment were pooled together and protein-wise linear models combined with empirical Bayesian statistics were used to identify differentially expressed proteins between bait versus non-treated control (FDR-adjusted $p$-values ≤ 0.05). Subcellular localization and enrichment were assessed using SubCellViz111 and hypergeometric tests, identifying significantly enriched compartments associated with receptor-mediated endocytic trafficking by identified prey proteins (Supplementary Table 2). All analyses were performed by investigators blinded to experimental conditions/groups.

### Protein enrichment of specific cellular compartments and pull-down protein interaction analysis

To evaluate C1q effect on total p32 protein or p32 protein availability on hNSC surface, the cell surface proteins were enriched using non-cell membrane permeable EZ-Link™Sulfo-NHS-SS-Biotin (ThermoScientific, 21331) and pull-down was performed using (Thermo Scientific, Streptavidin Agarose Resin, 20347). Extraction of cytoplasmic, membrane/structural, nuclear, and insoluble proteins from hNSC was performed using a sequential detergent- and salt-based subcellular fractionation. Analyses were performed as a single biologically independent experiment to validate the Imagestream and immunocytochemistry data (Supplementary fig. 4p, q). Step-by-step protocols describing the procedures used in this study for have been deposited on protocols.io and can accessed as follows: Enrichment of surface proteins using biotin dx.doi.org/10.17504/protocols.io.rm7vz9qz4gx1/v1, and subcellular fractionation dx.doi.org/10.17504/protocols.io.q26g7n9o8lwz/v1.

Direct interactions among C1q, BAI1, and p32 were tested using His-tagged recombinant BAI1 or p32 (SinoBiological) as bait and purified or untagged C1q/p32 (MyBioSource, LSBio) as prey. Pull-downs were performed with Dynabeads His-Tag Isolation (Invitrogen, 10103D) in two biologically independent experiments. A step-by-step protocol describing in situ pull-down assay to test direct interaction between C1q, BAI1 and p32 has been deposited on protocols.io and can be accessed through: dx.doi.org/10.17504/protocols.io.kxygx4qjzl8j/v1. The original blots are available in the Source Data.

### Two-photon fluorescence lifetime microscopy and phasor data

NADH/NAD+ redox ratio indicating glycolytic and OXPHOS states in hNSC was measured using a label-free, phasor approach with two-photon FLIM as previously described[64–66]. hNSC were cultured overnight on PLO-laminin-coated 8-well coverslip slides and then treated with or without C1q for timecourse analysis. FLIM images were captured using a two-photon microscope with a Ti:Sapphire laser (Spectra-Physics Mai Tai) and a Zeiss Axiovert S100TV 40x objective at 740 nm. A dichroic filter (700DCSPXR, Chroma Technologies) a hybrid detector (HPM-100, Hamamatsu) was used for separating and detecting the signal. Fluorescein lifetime was used for calibrating the detection, and the data was processed using SimFCS software (UCI Laboratory for Fluorescence Dynamics). Data from 5-9 cells over 3 biologically independent experiments were analyzed and phasor plot values calculated for each cell. All analyses were performed by investigators blinded to experimental conditions/groups.

### Mitochondria segmentation and morphology

hNSC were cultured on PLO-laminin-coated 8-well slides in GM overnight. The next day, cells were treated with 200 nM C1q or equivalent amount of GM (non-treated control) for 24 h. Cells were washed and stained with 100 nM MitoTracker™ Red CMXRos (Invitrogen, M7512) for 30 min. Live cell imaging was performed in pre-warmed GM using a

Zeiss LSM900 Airyscan with a 60x objective and z-stacks of 0.5 μm optical sections. Mitochondria were segmented and analyzed with CellProfiler version 4.2.1 (CellProfiler) and MitoMo (https://www.vislab.ucr.edu/SOFTWARE/software.php) followed by Matlab (MathWorks) analysis as previously described[108]. All biologically independent experiments were performed with technical duplicates. Analyses were performed by investigators blinded to experimental conditions/groups. Values of each independent experiment were tested for outliers using Grubb's outlier test (Alpha 0.2) in Prism; outliers are listed in the Source Data.

### Seahorse mitostress test
hNSC were seeded at 40,000 cells/well on PLO-laminin-coated XF24 microplates (Agilent Technologies). The next day, cells were treated with 200 nM C1q or equivalent amount of GM for 48 h. Metabolic status was assessed using the Seahorse XF24 Cell Mito Stress Test Kit and Seahorse XF24 extracellular flux analyzer (Agilent Technologies) according to the manufacturer instructions. More details available in the Supplementary Methods. Data were analyzed Wave version 2.6.1 (Agilent Technologies) and normalized to total protein of respective sample wells measured using Qubit (Thermo Fisher Scientific, Q33211). All biologically independent experiments were performed with at least technical triplicates. Wells with technical errors were excluded from the analysis. Basal OCR and ECAR are means of three basal reads per group per each independent experiment. Normalized values for each experiment were tested for outliers using Grubb's outlier test (Alpha 0.05) in Prism; exclusions and outliers are listed in the Source Data.

### Spinal cord injury, stem cell transplantation, and histological analysis
10-week-old Rag-1-immunodeficient female mice (Jackson Laboratories, B6;129S7-Rag1tm1Mom/J, #002096) ($n = 14$) received a moderate 60kD contusion injury at vertebral T9 under anesthesia as previously described in refs. 109–112. Immediately after injury, mice received four 250 nL injections (total volume 1 μL; 75,000 cells) of either BAI1 WT or BAI1 KO hNSC into spinal cord intact parenchyma adjacent to lesion epicenter as previously described in refs. 109–112. All mice received standard post-operative care, including Ringer's lactate solution (MWI Animal Health), Bubrenorphine (MWI Animal Health), Baytril (Patterson Veterinary) SQ, and twice a day bladder care. To label proliferating cells, mice were given 50 mg/kg BrdU i.p. twice a day until the end of the study 2 week post-injury/transplant (wpt).

For histological analysis, mice were anesthetized and perfused with 4% PFA at 2wpt, and T6–T12 spinal cord segment was dissected, post-fixed in 20% sucrose/4% PFA, and frozen in isopentane at −65 °C. Transverse sections were cut at 30 μm using a CryoJane tape transfer system (Leica Microsystems). Sections in the 1/12 sampling sequence were processed for antigen retrieval using a citrate buffer (PickCell Laboratories, Retriever 2100). Antibodies and dilutions used were human-specific STEM121 (1:2000, Takara, AB281314) and BrdU (1:200, Abcam, Ab1893) antibodies. Unbiased stereological analysis was conducted on an Olympus IX51 microscope with a 60x objective using systematic random sampling, an optical fractionator probe, and StereoInvestigator version 9 (MicroBrightField) as previously described in refs. 111,112. The sampling was designed to meet stereological validation principles. Animal care and analysis were performed by investigators blinded to experimental conditions/groups.

### Statistics and reproducibility
Statistical analysis methodology including corresponding details can be found both within the Figure Legends and in the Source Data. All data are presented as mean ± standard mean of error (s.e.m). Number of biologically independent replicates or measurements referring to numbers each experiment was repeated independently with similar results are listed in the Figure legends and in the Source Data. Before conducting statistical analyses, the data were assessed for normality and homogeneity of variances, and appropriate parametric or non-parametric tests were applied as outlined in the Source Data or Supplementary Source Data.

### Ethics statement
All human stem cell experiments were approved by the Human Stem Cell Research Oversight committee (#3682) and Institutional Biosafety committee (BUA-R120) at UCI. Informed consent was obtained from all fetal tissue donors. Mouse procedures and mNSC derivation followed UCI Institutional Animal Care and Use Committee (IACUC; AUP-23-023) and NIH animal care guidelines.

### Reporting summary
Further information on research design is available in the Nature Portfolio Reporting Summary linked to this article.

## Data availability
Source Data are provided with this paper. These data have been deposited in the FigShare database and are available as follows: Source Data https://doi.org/10.6084/m9.figshare.30413113.v1 and Supplementary Source Data https://doi.org/10.6084/m9.figshare.30413785.v1. The raw proteomic and RNAseq data are protected and are not available due to data privacy laws. The Sangers sequencing as well as processed proteomic and RNAseq data have been deposited in the FigShare database and are available as follows: Sangers sequencing [https://doi.org/10.6084/m9.figshare.29802875.v1]; Processed RNAseq [https://doi.org/10.6084/m9.figshare.29803526.v1]; and processed protein MS data [https://doi.org/10.6084/m9.figshare.29803532.v1]. BAI1 WT and KO hNSC lines are available from the lead contact upon request.

## Code availability
This paper does not include any original code.

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

## Acknowledgements

The authors are grateful to: Rebecca Nishi, MS for assistance with SCI surgeries; Jennifer Atwood, Dr. Eric Pearlman and the Institute for Immunology Flow Core for assistance with Imagestream; Vanessa Scarfone, Pauline Nguyen and the UCI SCRC Flow Core for assistance with cell sorting; the UCI Genomics High Throughput Facility for RNA-seq; Marilyn Chwa and Dr. Christina Kenney for assistance with Seahorse analysis; Dr. Enrico Gratton and the Laboratory for Fluorescence Dynamics for FLIM; Drs. Lan Huang, Clinton Yu, and the UCI High-End Mass Spectrometry Facility for LC-MS/MS analysis; Chunni Zhu, Marianne Cilluffo, the UCLA Microscopic Techniques Laboratory, and the UCLA Electron Microscopy Laboratory for TEM imaging. We also express our appreciation to Chris Nelson, Krystal Godding, BS, and Joseph Requejo, BS, for technical assistance, and to Ara Salibian, MD, Eileen Procanik, MS, Colleen Stone, BS, Kulbir Singh, BS, Olivia Tsai, MD, Xinyue Pei, MS, Yuqian Wang, MS, Taylor Nakayama, BS, Naomi Imbre, BS, Yuriy Maksymyuk, MD, Kevin Yang, BS, and Bryant Le, BS, for imaging and data analysis. This work was supported by Wings for Life funding (WFL-US-01/18), the Johnson Family Gift, and R01NC123927 to A.J.A., and the California Institute for Regenerative Medicine (CIRM) Bridges to Stem Cell Research Program grants EDUC2-12638 to V.N. and A.O.M.-P., EDUC2-08383 to W.A.S. and A.P., and EDUC2-08418 to A.N.R.

## Author contributions

Conceptualization: K.M.P and A.J.A.; Methodology: K.M.P, A.L., F.B-P., Z.H.E., A.A.N., N.T.A, A.Z., A.A., C.M.C., and A.J.A.; Investigation: K.M.P, A.L., F.B-P., Z.E., A.A.N, N.T.A., V.N., W.A.S., X.C., J.D., A.P., A.O.M-P., A.N.R., A.A, and C.M.C.; Visualization: K.M.P., A.L., F.B-P., Z.H.E., A.A.N., N.T.A., V.N., A.Z., W.A.S, J.D., A.P, A.O.M-P., A.N.R., A.A., and C.M.C.; Supervision: K.M.P, C.M.C., A.J.A.; Project Administration: K.M.P.; Funding Acquisition: A.J.A.; Writing–Original Draft: K.M.P. and A.J.A.

## Competing interests

The authors declare no competing interests.
