## [Transparent Peer Review file · Nature Communications]

C1q drives neural stem cell quiescence by regulating cell cycle and metabolism through BAI1

Corresponding Author: Dr Katja Piltti

Version 0:

Reviewer comments:

Reviewer #1

(Remarks to the Author)

This manuscript reports effects of C1q on human neural stem cells (hNSCs) mediated by the receptor BAI1. The authors describe effects of C1q on hNSC proliferation and other aspects of the cells' physiology. In parallel studies, the authors provide insight into the signaling pathways that are engaged when C1q binds BAI1, specifically i) regulation of mdm2 to control p53 and ii) endocytosis of C1q to modulate p32. These findings are novel, although the connections between the signaling pathways elucidated and the effects of C1q on hNSC proliferation are more hypothetical than directly shown. Specific comments are as follows:

1. The central physiological effect reported by the authors is a striking effect of C1q on hNSC proliferation (Fig. 1a). The authors also provide evidence that this effect is mediated by C1q engagement of BAI1 (Fig. 6). Separately, the authors provide evidence that C1q can regulate p53 levels in hNSCs (Fig. 3) and additionally regulate p32 and aerobic glycolysis after gaining entry into hNSCs via BAI1 (Fig. 4-5). However, it is not clear from the data presented whether these mechanisms actually account for the striking effect of C1q on hNSC proliferation shown in Figure 1. To put it another way: Figs. 1 & 6 show that C1q regulates hNSC proliferation in a BAI1-dependent manner, and the data from Fig. 3-5 provide insight into various signaling pathways that are activated when C1q engages BAI1 in hNSCs. What's not shown, though, is whether these signaling pathways are directly responsible for the effect of C1q on hNSC proliferation. Given that BAI1 is known to couple to other signaling pathways (Rac, Rho, etc) that are capable of affecting proliferation, the authors need to support their proposed model by more definitively demonstrating that the mechanisms they describe (regulation of p53 and p32) are in fact responsible for directly mediating the ability of C1q to regulate hNSC proliferation.

2. In creating the BAI1 KO cell line, the authors show the proper DNA fragment sizes (Extended Data, Figure 2). However, in the Western blot data shown in Fig. 3b, there are still BAI1-positive bands in the BAI1 KO lanes, which means that BAI1 may not actually be deleted at the protein level. The authors should address this point. If the authors believe these bands in the KO lanes to be non-specific, they should state what evidence leads them to this conclusion. This is an important issue to resolve, given that many of the authors' conclusions depend on BAI1 being fully deleted from the KO cell line.

3. For the experiments with the BAI1 neutralizing antibody (nAb) shown in Fig. 6, it is unclear which BAI1 antibody was used by the authors for these experiments. Also, how was this antibody characterized? How do the authors know that it is truly a neutralizing antibody? Obviously, it is possible for antibodies to bind to a receptor but not neutralize ligand binding, and thus it is important for the authors to provide more information about the characterization of this antibody. The authors provide a citation for a previous characterization of the C1q nAb that was used in these studies, but no comparable citation is provided for the BAI1 nAb.

Reviewer #2

(Remarks to the Author)

Review of the manuscript entitled "C1q drives neural stem cell quiescence by regulating cell cycle and metabolism through BAI1".

In this manuscript, the authors describe a role for the complement-derived pattern recognition molecule C1q in neural stem

cell (NSC) quiescence. The team had previously identified several new receptors for C1q, including brain angiogenesis inhibitor (BAI1). Here, they follow up this functional interaction and demonstrate C1q binding to BAI1 on NSCs induces, at minimum, two events: firstly, BAI1-mediated sequestering of MDM2 which, in turn, increases p53 phosphorylation and blocks cell cycle progression and secondly, BAI1-dependent internalization of C1q which enables interaction of C1q with the mitochondrial C1q receptor p32. The latter then induces a shift from OXPHOS to aerobic glycolysis which also supports quiescence. As a general model, the authors argue that age- or inflammation/trauma induced increases in local C1q production may sustain NSC quiescence and inhibit activation when desired (during tissue repair, etc.). The team makes its case using a set of elegant and detailed in vitro experiments and argues that this pathway is important in vivo by showing that the engraftment of BAI-KO human NSC into the brains of mice exposed to an acute CNS injury model (which is associated with local increases in C1q) rescues the decline in hNSC proliferation when compared to WT hNSC.

Overall, the study is interesting and timely and its strength lies in the fact that it further explored a new C1q-receptor interaction. Given the emerging central role for C1q across human disease and specifically in CNS pathology settings, C1q is becoming a valuable therapeutic target. The large majority in the field, however, focusses on the 'canonical' C1q-receptor interactions (gC1qR and cC1qR). C1q biology is highly complex, and it is therefore paramount to better understand its biology for successful future targeted interventions. This leads to the second strength of the work: the authors argue that the effects of C1q in each tissue will depend on the C1q receptor 'availability' which then also dictates the specific outcome of C1q activity. This aligns well with the emerging realization that local complement impact is highly cell and context specific.

There are several areas where the work can (and needs to be improved):

1. A major issue is that the authors have not satisfactorily dissected the effects on proliferation, self-renewal, cell death and survival. Particularly Extended data Figure 3B and C is difficult to reconcile as the control phenotype seems to be driven by a single cell that divides extensively.
2. Although the authors likely want to make it easier 'on the eye' by showing the control data as a 100% threshold line, raw data are preferred to see the spread here.
3. In general, all experiment dealing with phosphorylated proteins (p53 etc.) need to show total protein as comparator).
4. The mitochondrial staining for C1q/p32 etc. are not convincing and should be repeated in high resolution (like Figure 5I, which is very nice).
5. The authors argue that BAL1 induces C1q internalization which then can bind to p32 as it may be released. However, there should be a high-resolution triple staining for C1q/BAL1/p32 (which should be negative) to support this model.
6. Figure 1E: something is not right here with the last 2 conditions: why does the isotype control IgG antibody further amplify the reducing effect of C1q on NSC proliferation and also has such a strong inhibitory effect without C1q addition?
7. Does ERK inhibition have similar effects on NSC proliferation?
8. Figure 4K: the negative C3aR/C1q control looks positive (?) or has a very high background which makes it difficult to interpret the other data – thoughts?
9. Figure 7: the analysis of C1q is very nice – why did the authors not perform the same analysis for BAI1? It would be nice to see the expression profile of this C1q receptor in relation to its ligand. It would have also added some suggestions to whether the interaction by be autocrine or paracrine.
10. The in vivo experiment in Figure 7 solidifies the role of BAL1 in hNSC proliferation control. It does not conclusively allow to associate this with the C1q activity though. Could the authors approach this somehow? For example, could there be a bulk RNA-seq analysis of in vitro WT and BAL1 KO hNST +/- C1q (as in Figure 6) to generate C1q-BAL1 axis (normal/perturbation) signatures and then utilize these signatures to assess the bulk RNA-seq signatures from re-isolated SC121-BrdU+ cells WT/KO cells in the lesions (Figure 7D)?
11. The manuscript has many errors when it comes to labeling and cross-referencing figures and data. For example, the table is references erroneously as Extended Data 2, there is no reference to Extended Data 3 in the text and Extended Data 8 (mentioned in the text) are entirely missing, Extended Data 7 legend seems not to fit (ROS), etc.. Overall, the structure of text, figures, and Extended data can be improved – it was at times difficult to navigate text and figures.

Reviewer #3

(Remarks to the Author)

In this work the authors focus on the complement factor C1q and its effects on proliferation and quiescence of human neural stem cells (hNSCs). First, they demonstrate that C1q treatment leads to these cells proliferating significantly less with the absence of any changes in apoptosis, suggesting an induction of quiescence. Next, they undertake a variety of studies to assess the underlying mechanism of this effect. The first of these mechanisms is shown through changes in p53 phosphorylation and localization to the nucleus which seems to occur through MDM2 initiated by binding to BAI1. They then assess the internalization of C1q and suggest that this occurs via endocytosis. The authors also detail a novel role for C1q in mitochondrial changes in hNSCs acting through p32 and BAI1. Finally, the authors show that loss of BAI1 rescues the C1q impairments in proliferation and that transplantation of BAI1 KO cells are more highly proliferative in a spinal cord injury model. These data are exciting and expand our understanding of the role of immune molecules in the control of proliferation and quiescence of neural stem cell populations. While I am enthusiastic about these studies, I do have a number of criticisms that the authors should consider.

-In Figure 1E, it is shown that IgG nab both alone or in the presence of C1q leads to a ~5-fold larger reduction in BrdU positive cells than when C1q is used alone. How can this be as this is a control antibody? Moreover, in this same figure it's shown that after priming for 4-8 weeks this actually leads to greater proliferation (ie. BrdU+ cells) than control. Can the

authors explain why this might be? Are the spheres just much larger but there are many fold less of them?

-In a number of cases, western blots are shown with bands that come from a number of different blots including when trying to demonstrate quantitative differences (for example Figure 1K). In such cases, these data need to come from one blot.

-In multiple figures (for example, Figure 2c and Figure 3d) no control images are shown and only treatment examples are used. This makes interpretation of the data difficult. In the case of Figure 2c, there is an apparent 4-fold increase in the presence of C1q which should be striking in images comparing Control to 200 nM C1q.

-While the BAI1 KO and C1q dependence in this work is convincing the lack of a direct demonstration of C1q and BAI1 interaction is a significant weakness. Considering that BAI1 recombinant protein is commercially available it should be straightforward to demonstrate a direct interaction using a technique such as biolayer interferometry or surface plasmon resonance or a more traditional pull-down assay. It remains possible that the interaction between BAI1 and C1q is indirect.

-The model with which p32 is involved with C1q and BAI1 doesn't seem fully plausible. The author's own data suggests that p32 is in the mitochondrial matrix while C1q in large part enters the cell via endosomes. How can these two proteins specifically interact when they are in different cellular compartments? This would require demonstrating that both C1q are in the same place at the same time in hNSCs at high resolution. Moreover, if the authors propose, as in Figure 5e, that C1q enters endosomes and then is released from endosomes this would need to be demonstrated experimentally.

Minor:

-Page 2 Line 1 BCB not defined. Perhaps BSB typo?

-In a number of cases "Blood-derived C1q concentrations" is used. This is confusing as the data shown here actually has nothing to do with "Blood-derived C1q." Consider modifying this terminology for clarity.

-In Figure 3B, this is the only blot shown as white bands on black background when all of the rest of the blots are shown inverted. This is confusing and can lead to confusion over whether these data are western or an agarose gel.

-In Figure 4i, pull down based proteomics data is introduced but these data are hardly described or mentioned at all and only shown as a cartoon. If these data are significant enough to be shown in the main figures they should be described and interpreted in a more fulsome manner.

Reviewer #4

(Remarks to the Author)

"I co-reviewed this manuscript with one of the reviewers who provided the listed reports. This is part of the Nature Communications initiative to facilitate training in peer review and to provide appropriate recognition for Early Career Researchers who co-review manuscripts."

Version 1:

Reviewer comments:

Reviewer #1

(Remarks to the Author)

This manuscript reports effects of C1q on human neural stem cells (hNSCs) mediated by the receptor BAI1. The authors describe effects of C1q on hNSC proliferation and other aspects of the cells' physiology. In parallel studies, the authors provide insight into the signaling pathways that are engaged when C1q binds BAI1, specifically i) regulation of mdm2 to control p53 and ii) endocytosis of C1q to modulate p32. These findings are novel and interesting.

The authors were very responsive to the reviewers' comments and have addressed all of the reviewers' concerns following the initial review. The textual changes and new data added by the authors have significantly strengthened the manuscript.

Reviewer #2

(Remarks to the Author)

The authors have addressed to the largest extent the queries and concerns raised by this reviewer satisfactorily. The only 'negative' remaining would be that the authors have missed an opportunity pinpoint then intracellular location of the C1q/p32 location better - this was also raised by reviewer 3. I am not sure why as this would have elevated to the work substantially and performing higher resolution/quality staining of mitochondria as suggested by this reviewer (or endosomal compartments as indicated by the authors) is not difficult. However, this reviewer does not feel sufficiently strongly about this matter to ask for another revision.

Reviewer #3

(Remarks to the Author)

The authors have suitably addressed a number of my concerns and the manuscript has improved substantially. However, a couple of points remain:

1) I congratulate the authors for carrying out the pull down experiment using each of the interacting proteins. These data are now in Figure 5F. The manner, however, in which these data are presented makes it nearly impossible to interpret. For example, it appears that BAI1 is used as the bait in all lanes (except 3 and 4) shown in these blots and the probed with antibodies above? The presentation of these data could be significantly improved by indicating bait and prey in each case. It took this reviewer, a significant amount of time reviewing the methods and looking at the figure and legend to get an idea of what is going on with that data. The data should be interpretable by simply looking at the figure and legend. In addition, in lanes 3 and 4, it appears that no BAI1 bait was used and yet C1q and p32 are pulled down. How can that be if only BAI1 has the his-tag to bind to the magnetic beads? Is this result of non-specific binding of C1q/p32 to the beads or something else? If the former, this would seriously impact these data and call them into question.

2) Related to the above, the authors indicate in a number of instances in the response that "The full original blots are in the Source data." This reviewer looked through every file in the manuscript tracking system and could not locate any original blots. These blots would help to interpret the data mentioned in point #1 above.

Reviewer #4

(Remarks to the Author)

Version 2:

Reviewer comments:

Reviewer #3

(Remarks to the Author)

The authors have suitably addressed my remaining concerns.

Authors response to Reviewers

We appreciate the valuable feedback on our manuscript, and the time and effort invested in providing detailed comments and suggestions. We have carefully considered each comment and made revisions accordingly. Additionally, as requested, all of the figures have been formatted to match Nature Communications requirements, and the source data files with raw data have been included in this resubmission. Below, we address each of the points raised.

Reviewer #1 (Remarks to the Author):

This manuscript reports effects of C1q on human neural stem cells (hNSCs) mediated by the receptor BAI1. The authors describe effects of C1q on hNSC proliferation and other aspects of the cells' physiology. In parallel studies, the authors provide insight into the signaling pathways that are engaged when C1q binds BAI1, specifically i) regulation of mdm2 to control p53 and ii) endocytosis of C1q to modulate p32. These findings are novel, although the connections between the signaling pathways elucidated and the effects of C1q on hNSC proliferation are more hypothetical than directly shown. Specific comments are as follows:

- 1. The central physiological effect reported by the authors is a striking effect of C1q on hNSC proliferation (Fig. 1a). The authors also provide evidence that this effect is mediated by C1q engagement of BAI1 (Fig. 6). Separately, the authors provide evidence that C1q can regulate p53 levels in hNSCs (Fig. 3) and additionally regulate p32 and aerobic glycolysis after gaining entry into hNSCs via BAI1 (Fig. 4-5). However, it is not clear from the data presented whether these mechanisms actually account for the striking effect of C1q on hNSC proliferation shown in Figure 1. To put it another way: Figs. 1 & 6 show that C1q regulates hNSC proliferation in a BAI1-dependent manner, and the data from Fig. 3-5 provide insight into various signaling pathways that are activated when C1q engages BAI1 in hNSCs. What's not shown, though, is whether these signaling pathways are directly responsible for the effect of C1q on hNSC proliferation. Given that BAI1 is known to couple to other signaling pathways (Rac, Rho, etc) that are capable of affecting proliferation, the authors need to support their proposed model by more definitively demonstrating that the mechanisms they describe (regulation of p53 and p32) are in fact responsible for directly mediating the ability of C1q to regulate hNSC proliferation.*

We have previously published that multiple intracellular signaling pathways contribute to C1q-driven proliferation in hNSC (PMID: 32894219). However, in our phosphoarray data, the only unique sustained signaling pathway in C1q [200nM]-treated hNSC was an increase in phosphorylation of p53 serine 15 (p53Ser15) (Fig.2a; PMID: 32894219); due to the previously published link between p53 and BAI1 in tumor cell proliferation (PMID: 29894688), we therefore focused on testing the effect and mechanisms of C1q-BAI1 interaction in hNSC proliferation. We appreciate Reviewer's interest in extending the identification of a novel role for C1q-BAI1 interaction in hNSC proliferation to include

establishment of a direct role for p53 and p32 signaling in this mechanism, however, attempts to do so have proven impossible.

Most critically, in our paradigm, disruption of either of p32 or p53 signaling leads to rapid cell death in hNSC. This is consistent with reports that p32 KO in mammalian systems is embryonic lethal before the peak of neurogenesis (PMID: 22904065); thus, analysis of the direct roles of p32 would require use of non-mammalian model systems (PMID: 36995025). Towards that goal, we attempted to establish a neural cell paradigm in *Drosophila*, in which p32 knockout has been reported, but were unable to produce viable cells. We also attempted to manipulate p53 signaling using the selective chemical inhibitor UC2288 (Sigma-Aldrich, 532813), which targets the downstream effector p21 (PMID: 23298903). However, consistent with the literature (PMID: 27156098), p53 pathway inhibition with UC2288 during C1q-treatment of hNSC induced apoptosis (Response letter_Fig.1). The negative impact of p32 and p53 signaling inhibition on dividing cells is not surprising, due to the energy requirements of these cells and role of p32 in mitochondrial function, as well as the expected effect of disrupting the cell cycle in a dividing cell. As a result, with the current tools available, there does not appear to be a viable approach by which to reasonably be able to address this review comment.

Finally, we would like to highlight that, as we discuss in the manuscript (page 8, lines 38-42 and page 9, lines 1-3), we do not propose that hNSC proliferation is uniquely modulated by C1q-BAI1 interaction, and predict that other putative C1q receptors may play roles in regulating this crucial biological process. Hence, while dissecting potentially converging signaling pathways may ultimately be required, we suggest that the fundamental novelty of the current findings remains an important contribution to the literature.

Response letter_Fig.1. Selective chemical inhibitor UC2288 targeting p53 downstream effector p21 increases hNSC death (CC3+ cells) relative to differentiation medium (DM) + DMSO controls (dash line) in C1q 200nM-treated hNSC.

2. In creating the BAI1 KO cell line, the authors show the proper DNA fragment sizes (Extended Data, Figure 2). However, in the Western blot data shown in Fig. 3b, there are still BAI1-positive bands in the BAI1 KO lanes, which means that BAI1 may not actually be deleted at the protein level. The authors should address this point. If the authors believe these bands in the KO lanes to be non-specific, they should state what evidence leads them to this conclusion. This is an important issue to resolve, given that many of the authors' conclusions depend on BAI1 being fully deleted from the KO cell line.

Thank you for your detailed comments regarding our hNSC BAI1 knockout (KO) line – we sincerely appreciate the opportunity to address these concerns. To clarify, at the end of our two-step genome editing strategy we created a precise genetic KO of the BAI1 gene in hNSCs by inserting an in-frame loss-of-function mutational scar (consisting of stop codons) into the start of the BAI1 open reading frame at amino acid 52 of 1584 total (96.7% size reduction). In this manuscript we present three lines of orthogonal evidence that demonstrate the successful silencing of BAI1 in our KO line as discussed below.

First, at the genetic-level, if the editing was successful, it is expected that the mutational scar at the BAI1 loci would increase the size of amplicons around the editing site by 44BP in KO cells only in comparison to the unedited alleles of wild-type (WT) cells. In Supplementary Fig. 2, we clearly demonstrate that we generated precise genetic BAI1 KO cells via sanger sequencing – providing the first line of evidence.

Second we believe we may have created confusion by including both reducing and non-reducing gel images in the main figures, and have refined the figure accordingly. All blots remain in the source data. While reducing gels will show changes in size because of disruption of disulfide bonds, non-reducing gels maintain the intact protein structure and therefore identify whether or not there is translation of the full protein size. As shown in Fig.3b (updated to a white background to match the style in rest of Western blot figures), in non-reducing western blots of WT and KO hNSC using anti-BAI1 antibody, WT cells show a primary band at 174kDa and two lighter bands below, and KO cells show elimination of the 174kDa band, with two barely visible bands below. The bands that appear below 174 kDa may reflect either post-translational modifications [PMID 38941066], or minor non-specific antibody binding under these conditions. However, most critically, the 174kDa full length protein band is completely absent in KO cells. Importantly, the consistency of the actin loading control across all samples rules out differences in protein loading or transfer efficiency in detection. We would also highlight that this human BAI1 antibody (AF4969; R&D systems) in Fig. 3b recognizes the 174kDa BAI1 band in BAI1 overexpression HEK293T cell lysates (+) (Response letter_Fig. 2). Note that in BAI1 overexpressing HEK293T cells, full-length BAI1 is observed at two molecular weights due to post translational modifications, however, the manufacturer reports <2% cross-reactivity with recombinant human BAI3 (170-130 kDa molecular weight). We have added the empty vector (-) negative HEK293T cell control lysates into the Supplementary Fig.3a.

Third at the functional-level, as shown in Fig. 3c-d, we further authenticate our BAI1 KO cells by performing PLA specific for the protein-protein interaction of BAI1 with its ligand, C1q, in both BAI1 KO and WT cells. PLA is capable of detecting protein-protein interactions with high specificity by generating a fluorescent signal only when antibodies bound to target proteins are in close proximity. In these experiments, our results clearly demonstrate the absence of fluorescent signal only in BAI1 KO cells upon C1q stimulation -- while in BAI1 WT cells, upon C1q stimulation, a significant increase in fluorescent signal is produced. These BAI1-C1q PLA results visually demonstrate that our BAI1 KO cells exhibit loss of BAI1 protein and function, further corroborating that BAI1 is fully deleted at the protein level in our BAI1 KO cell line.

Taken together, these genetic, proteomic, and functional data support that BAI1 expression has been effectively deleted in KO hNSC

We have updated the Figure legends, Methods, and Online Methods to include the additional information as shown below:

“Fig.3: **b**, Loss of full-length 174kDa BAI1 protein in BAI1 KO vs. WT hNSC line visualized by Western blot.”

“Supplementary Fig.3: **a**, Commercial BAI1 antibodies generated using peptides from C-terminal vs. N-terminal region recognize full-length 174kDa BAI1 protein when analyzed using BAI1 overexpression lysate (+) vs. empty vector negative HEK293T control lysate (-) (Novus Biologicals) via Western blot under reducing conditions.”

“Methods: CRISPR Cas9 hNSC BAI1 genetic deletion

Page 14, line 40 forward: Stable loss of total BAI1 protein (174 kDa) expression was confirmed by Western blotting (Fig.3b), and functional-level, by performing PLA specific for the protein-protein interaction of BAI1 with its ligand, C1q, in both BAI1 KO and WT cells (Fig.3c-d). Two different BAI1 KO lines were generated with the same method. No substantial differences between the cell lines generated were detected regarding their CD133+ content, multipotency, migration, or proliferative capacity. Follow-up experiments *in vitro* were done at least independent experimental triplicates. BAI1 WT and BAI1 KO hNSC exhibited sustained normal karyotype (analyzed by Cell Line Genetics), high CD133+ stem cell proportions, stable growth rate under *in vitro* growth conditions, migration response, and

Response letter_Fig. 2. BAI1 band in BAI1 overexpression HEK293T control (+) vs. HEK293T empty vector negative control (-) lysate.

multipotency in neural lineage differentiation (Supplementary Fig.2f-n) demonstrating that both cell lines retained normal hNSC characteristics.”

“Online Methods: BAI1 KO hNSC characterization

Page 29, line 337 forward: Human BAI1 antibody (R&D systems, AF4969) used for analyzing total BAI1 protein in BAI1 WT vs. KO hNSC in non-reducing Western blots (Fig. 3b) recognizes the correct BAI1 band (174 kDa) in BAI1 overexpression HEK293T cell lysate (+) (Novus Biologicals, NBP2-08183) (Supplementary Fig. 3a). In BAI1 overexpressing HEK293T cells, full-length BAI1 is present at two different molecular weights due to post translational modifications. Manufacturer of the antibody reports <2% cross-reactivity with recombinant human BAI3 at 170-130 kDa molecular weight.”

3. For the experiments with the BAI1 neutralizing antibody (nAb) shown in Fig. 6, it is unclear which BAI1 antibody was used by the authors for these experiments. Also, how was this antibody characterized? How do the authors know that it is truly a neutralizing antibody? Obviously, it is possible for antibodies to bind to a receptor but not neutralize ligand binding, and thus it is important for the authors to provide more information about the characterization of this antibody. The authors provide a citation for a previous characterization of the C1q nAb that was used in these studies, but no comparable citation is provided for the BAI1 nAb.

Thank you for bringing this to our attention. For BAI1 neutralization we used Human BAI1 Antibody (Fig. 6a; R&D systems, AF4969). Data from the manufacturer demonstrates the capacity of the BAI1 antibody to neutralize the the enhancement of adhesion of BCE C/D1b bovine cornea derived endothelial cells induced by BAI-1 at a dose of 2-8 $\mu\text{g}/\text{mL}$ in the presence of 0.5 $\mu\text{g}/\text{mL}$ Recombinant Human BAI-1. Additionally, characterization of cross reactivity in direct ELISAs identifies less than 2% cross-reactivity with recombinant human BAI3, supporting the specificity of this effect. We have added this information about the antibody in the Results Page 7, Line 7, Methods Page 11, Line 15, and the Online Methods Key Resources Table.

Reviewer #2 (Remarks to the Author):

Review of the manuscript entitled “C1q drives neural stem cell quiescence by regulating cell cycle and metabolism through BAI1”.

In this manuscript, the authors describe a role for the complement-derived pattern recognition molecule C1q in neural stem cell (NSC) quiescence. The team had previously identified several new receptors for C1q, including brain angiogenesis inhibitor (BAI1). Here, they follow up this functional interaction and demonstrate C1q binding to BAI1 on NSCs induces, at minimum, two events: firstly, BAI1-mediated sequestering of MDM2 which, in turn, increases p53 phosphorylation and blocks cell cycle progression and secondly, BAI1-dependent internalization of C1q which enables interaction of C1q with the mitochondrial C1q receptor p32. The latter then induces a

shift from OXPHOS to aerobic glycolysis which also supports quiescence. As a general model, the authors argue that age- or inflammation/trauma induced increases in local C1q production may sustain NSC quiescence and inhibit activation when desired (during tissue repair, etc.).

The team makes its case using a set of elegant and detailed in vitro experiments and argues that this pathway is important in vivo by showing that the engraftment of BAI-KO human NSC into the brains of mice exposed to an acute CNS injury model (which is associated with local increases in C1q) rescues the decline in hNSC proliferation when compared to WT hNSC.

Overall, the study is interesting and timely and its strength lays in the fact that it further explored a new C1q-receptor interaction. Given the emerging central role for C1q across human disease and specifically in CNS pathology settings, C1q is becoming a valuable therapeutic target. The large majority in the field, however, focusses on the 'canonical' C1q-receptor interactions (gC1qR and cC1qR). C1q biology is highly complex, and it is therefore paramount to better understand its biology for successful future targeted interventions. This leads to the second strength of the work: the authors argue that the effects of C1q in each tissue will depend on the C1q receptor 'availability' which then also dictates the specific outcome of C1q activity. This aligns well with the emerging realization that local complement impact is highly cell and context specific.

There are several areas where the work can (and needs to be improved):

1. A major issue is that the authors have not satisfactorily dissected the effects on proliferation, self-renewal, cell death and survival. Particularly Extended data Figure 3B and C is difficult to reconcile as the control phenotype seems to be driven by a single cell that divides extensively.

We agree that understanding the complex biology of C1q biology important for future targeted interventions, and that these data combined with our previous description of C1q receptors provide insight the cell and context specific impacts of C1q. Indeed, we have several followup manuscripts addressing this in preparation, including one that focuses on the role of C1q in decline of neurogenesis in aging brain.

In interpreting this comment, we feel that we may have failed to provide sufficient clarity in the figure legends and methods to ensure that the data presented can be fully interpreted. In the response below, we address the Reviewer's concern regarding ***Supplementary Fig.3b and c*** and the contribution of a single cell to the control phenotype, as well as the underpinnings of how proliferation, self-renewal, cell death and survival.

First, we would like to note that ***Supplementary Fig. 3b,c*** shows supporting schematic examples for single cell Imagestream multispectral imaging flow cytometry gating for ligand internalization (Supplementary Fig.3b), receptor-ligand intracellular co-localization (Supplementary Fig.3c) and spot counts using IDEAS software, relevant for Imagestream data in Fig.4 and Fig.5. ***Accordingly, we believe the reviewer intended***

to refer to *Supplementary Fig.1d,e in the revised manuscript*, which is an example of the paradigm employed for quantification of cell death in the control vs. C1q-treated Fucci mNSC during fluorescent live-cell imaging (Fig.1j).

In ***Supplementary Fig.1d,e***, the 30 cells shown for each group represent a single biological replicate. However, a total of $n=4$ independent experiments ($n=120$ cells/group) were used to generate the data shown in the main figure (Fig. 1j), making the impact of any single cell on the final analysis quite small. In addition, it is important to consider what that impact would be; removing the cell in question would only serve to increase the proportion of dying cells in the control condition, thereby further enhancing the observed effect of C1q. Finally, as shown in *Supplementary Fig.1a,b* in the revised manuscript, we also report quantification of the effect of C1q at the same timepoint via immunocytochemical analysis of cleaved caspase-3 (CC3) staining, again conducted across 3 independent experiments, providing orthogonal validation of this result.

In addition to clarifying that number of replicates and cells quantified in *Supplementary Fig. 1d,e* are sufficient to enable interpretation, we have sought to clarify this in parallel for the other experiments that dissect the effects of C1q on proliferation, self-renewal, cell death and survival. As described in the manuscript, we performed a combination of analyses including BrdU/Edu proliferation assays, RealTime-Glo Annexin V multiplex assays, live cell imaging of Fucci-mNSC for cell cycle phases in subsequent timepoints and cell tracking for cell death, CC3 immunostaining, clonal neurosphere assays, RNAseq pathway enrichment analysis of quiescence, activation, and neural differentiation associated transcripts, and Western blot analysis of total protein for undifferentiated NSC/progenitor markers vs. lineage commitment markers. In addition, the effects of C1q on NSC were also tested after short-term priming (up to 48h; proliferation) and long-term priming (2, 4, 6, or 8 weeks; proliferation, self-renewal, cell death, differentiation) to model prolonged C1q elevation caused by CNS injury/aging in order to dissect the effects of C1q on proliferation, self-renewal, cell death and survival. Accordingly, we have updated the Figure legends and Methods to include additional information as shown below:

“Fig 1.**h-i**, Fluorescent live-cell imaging-based quantification of G0/G1 (non-fluorescent/Cdt1-red) (**h**), or G1/S, S, G2 and M (Geminin-green) (**i**) cell cycle phase proportions in C1q-treated Fucci-mNSC vs. controls ($n = 4$ independent experiments). Data represent 150 cells plated/group per experiment, total $n=600$ cells plated/group based on quantification of all cells in each microwell at each timepoint. **j**, Survival of C1q-treated Fucci-mNSC during live-cell fluorescence imaging vs. controls quantified using single-cell tracking of randomly selected cells from 0d to 7d ($n = 4$ independent experiments, 30 cells/group per experiment, total $n=120$ cells/group; tracking example in *Supplementary Figs.1d,e*.”

“*Supplementary Fig.1: d-e*, A representative pedigree map of Fucci-mNSC live cell fluorescence imaging, in which 30 cells in G1 phase were randomly selected for tracking at 0h in each group. This is a graphic

example the data collection that was used for Fig.1j, in which n=4 independent experiments with 30 cells per replicate/group (a total of 120 cells per group) were used to test for differences in viability, comparing non-treated control (d) vs. C1q [200nM]-treated Fucci-mNSC over the course of a 7-day experiment (e). 30 cells in G1 phase were randomly selected for tracking at 0h in each group per independent biological experiment.”

“Methods: Page 11, Fluorescence live cell imaging and tracking

Fucci-mNSC were plated into custom-made PDMS microwells at a density of 150 cells/well without (non-treated control) or with 0.1nM or 200nM C1q in DM, imaged using Olympus Vivaview® fluorescence incubator microscope at 20x objective in 20min intervals for 7 days. Image analysis was performed using Imaris version 7.5.2. from image montages consisting of 3x3 stitched images as previously described²³. For each independent experiment five technical replicate microwells were seeded, and one microwell with comparable cell confluences per group was captured and analyzed. The total number of Fucci-mNSC and relative proportions of the cells in G1 or S, G2, or M phases in each timepoint per well were quantified from the image montages at 0h, 48h, 72h, 96h, and 7d during *in vitro* culture (Fig.1g-i, Supplementary Fig.1c).

For single-cell tracking of NSC survival (Fig. 1j, Supplementary Fig.1d,e), 30 Fucci-mNSC (all in G1 within the first frame of the movie corresponding to 0h timepoint) were randomly selected per each independent experiment to be tracked frame-by-frame for up to 7 days using the manual tracking feature of Imaris. Cell death of the tracked Fucci-mNSC is shown as an average percentage of dead cells relative total number of tracked cells (Fig. 1j). An example of the paradigm employed for quantification of viability in the Fucci-mNSC (Supplementary Fig.1d,e) represents a single independent experiment; 30 cells for each group. A total of 4 independent experiments (Total n=120 cells/group) was used to generate the data shown in Fig. 1j. All analyses were performed by investigators blinded to experimental conditions/groups.”

“Online Methods: Page 28, Quantification of cell cycle phases and cell survival during fluorescence live cell imaging

For cell cycle-, single cell division, death analysis, Fucci-mNSC were plated into custom-made PDMS microwells at a density of 150 cells/well without (non-treated control) or with 0.1nM or 200nM C1q in DM. For each independent experiment, five technical replicate microwells were seeded, and one microwell with comparable cell confluences per group was captured and analyzed. The total number of Fucci-mNSC and relative proportions of the cells in G1 or S, G2, or M phases per microwell in each timepoint were quantified from the image montages at 0h, 48h, 72h, 96h, and 7d during *in vitro* culture (Fig.1g-i, Supplementary Fig.1c). The data of

each independent experiment was collected from a single microwell on separate petri-dishes, that were either imaged separately (fluorescence time-lapse) or simultaneously (DIC-only time-lapse). A total of 4 independent experiments (n=150 cells plated for each group per each replicate) was used to generate the data shown in the main figure (Fig.1g-i, Supplementary Fig.1c) based on quantification of all cells in each microwell at each timepoint; (Total n= 600 cells plated/group).

For single-cell tracking of NSC survival (Fig.1j, Supplementary Fig.1d,e), we randomly selected 30 Fucci-mNSC (all in G1 within the first frame of the movie corresponding to 0h timepoint) per each independent experiment to be tracked frame-by-frame using the manual tracking feature of Imaris for up to 7 days; tracking experiments included the progeny of all selected parent cells. In each frame, a visually determined and manually selected physical center of the cell was used as a reference point for cell tracking. After cell division, both daughter cells were tracked until the end of the experiment unless the cell died or was lost to identification. An example of the paradigm employed for quantification of viability in the Fucci mNSC (Supplementary Fig.1d,e) represents a single independent experiment; 30 cells for each group.”

2. Although the authors likely want to make it easier ‘on the eye’ by showing the control data as a 100% threshold line, raw data are preferred to see the spread here.

We have included the Source data files with raw data per each independent experiment into this submission.

3. In general, all experiment dealing with phosphorylated proteins (p53 etc.) need to show total protein as comparator).

We thank the reviewer for raising this point. We have revised the Western blot data in Fig.2b and Fig.3i in the resubmitted manuscript accordingly; total p53^{Ser15} protein are now normalized to total p53 protein.

See also edits in “Methods: Page 12, line 33 ..anti-Phospho-p53 (Ser15) (1:300, Cell Signaling, 16G8, 9286), anti-p53 (1:500, Cell Signaling, 9282s)” and “Page 13, line 2: Phospho-p53 (Ser15) bands were normalized to total p53. “

“Fig.2: Blood plasma C1q concentration activates p53 signal transduction driving cell cycle quiescence. **b**, Western blot analysis of phospho-p53^{Ser15} in hNSC treated with C1q [200nM] vs. non-treated or heat-inactivated C1q [200nM] (T°Inact 200nM) negative controls. Optical band intensities were normalized to p53 and shown relative to non-treated controls (n = 4 independent experiments).”

“Fig.3: C1q-driven p53 signal transduction is BAI1-dependent. **g**, Western blot analysis of total phospho-p53^{Ser15} in C1q-treated BAI1 WT or KO hNSC relative to each non-treated control (ctrl, dashed line) (n = 4-5 independent experiments). Optical band densities were normalized to total p53.”

4. The mitochondrial staining for C1q/p32 etc. are not convincing and should be repeated in high resolution (like Figure 5I, which is very nice).

We thank the reviewer for bringing this to our attention.

We have interpreted this comment as best we can. We do not show mitochondrial staining for C1q/p32 in the original manuscript. We show mitochondrial staining in Fig 5I (now 5L), which is mitotracker at high resolution to enable quantification of mitochondrial segmentation/morphology. Elsewhere, we show staining for DAPI, mitotracker, and phospho-p53 in Figure 3J, where the point of including mitotracker was to identify cytoplasmic versus nuclear (DAPI) area, in order to quantify the nuclear translocation of phospho-p53. We have clarified this in the resubmission, and removed a graph of mitochondrial localization from this figure in deference to the reviewer’s comment. We do not feel that this use of mitotracker necessitates re-imaging at higher resolution. We believe that the other relevant data to this comment is in Supplementary Fig. 5e, which shows DAPI, mitochondrial marker TOM20, and p32 in C1q treated WT vs KO hNSC, with the principal goal to quantify p32 localization in the mitochondria. Again, we believe that because mitochondrial morphology is not analyzed here, the resolution is sufficient for this analysis.

We have revised the Resubmitted Fig.3 legend to correspond the data shown in the graph:

“C1q-driven p53 signal transduction is BAI1-dependent. **j**, Example images of phospho-p53^{Ser15} staining in Dapi+ nucleus vs. cytoplasm with Mitotracker+ mitochondria. **k**, Quantification of phospho-p53^{Ser15} localization in nucleus in C1q-treated BAI1 WT or KO hNSC using ICC relative to each controls (n = 3-4 independent experiments). Mean \pm s.e.m, **k** 1-way ANOVA with post hoc test.”

5. The authors argue that BAL1 induces C1q internalization which then can bind to p32 as it may be released. However, there should be a high-resolution triple staining for C1q/BAL1/p32 (which should be negative) to support this model.

The principal hypothesis we intended to test was whether internalized C1q interacts with intracellular p32 in a BAI1-dependent manner, as shown by PLA (Fig.4v-x; Fig.5c-d). In reflecting on the Reviewer’s question, we realized that the schematic in Figure 5G (hypothesized metabolic outcome for BAI1-dependent intracellular C1q-p32 interaction and decreased p32 stability) was misleading because - although not intended - it implied C1q release. As intracellular protein interactions and trafficking are areas where

much of the field of cell biology is seeking new understanding, the question of whether C1q is released from BAI1 within the endosomes remains to be determined and is outside of the scope of this submission. However, we appreciate the Reviewer’s point in requesting direct evidence for interactions after internalization, and have sought to provide this data. We have therefore added analyses of C1q-BAI1-p32 interactions using pull-downs of purified human recombinant proteins *in situ*. These data demonstrate direct interactions between C1q-BAI1, C1q-p32, and BAI1-p32, and that C1q-BAI1-p32 can be pulled down as a complex (Response letter_Fig. 3, Fig.5f). In deference to the Reviewer’s points, we have also revised the cartoon in Fig.5g to show C1q-BAI1-p32 in complex, and added a line to the discussion addressing the uncertainty of how dynamic this complex might be within the cell.

Response letter_Fig. 3. Direct interaction between C1q-BAI1, C1q-p32, and BAI1-p32 tested using *in situ* pull-down of human purified or recombinant proteins.

We have revised the Results, Figure legend, Discussion, and Methods accordingly:

“Results, Page 6, lines 8-12: Pull-down of purified human proteins *in situ* demonstrated a direct interaction between C1q-BAI1 and C1q-p32, as well as BAI1-p32, and that C1q-BAI1-p32 can be pulled down as a complex (Fig.5f). Together, these data show that BAI1-mediates endocytosis of C1q enabling intracellular protein interactions between C1q-p32, BAI1-p32, or C1q-BAI1-p32, decreasing total p32.”

“Fig.5: C1q-p32 intracellular interaction drives p32 instability and aerobic glycolysis in BAI1-dependent manner. **f**, Pull-down of human proteins *in situ* demonstrates direct interaction between C1q-BAI1, C1q-p32, and BAI1-p32.”

“Discussion: Page 9, lines 3-6: In addition, our data demonstrates a direct interaction not only between C1q-BAI1 and C1q-p32 but also BAI1-p32, enabling pulldown of C1q-BAI1-p32 as a complex. The role of BAI1-p32 interaction as well as how dynamic C1q-BAI1-p32 complex might be within the cell remains uncertain.”

“Methods: Page 19, Pulldown protein interaction analysis.

Direct interaction between C1q, BAI1 and p32 *in situ* was tested by pulldown assays using purified human proteins following the manufacturer instructions. Briefly, 5 µg bait His tag recombinant human BAI1 (SinoBiological, 4969-BA) was prepared in a 1x binding/wash buffer (50 mM sodium phosphate, 300 mM NaCl, 0.01% Tween-20) and immobilized on 2 mg magnetic beads (Dynabeads His-Tag Isolation and Pulldown, Invitrogen, 10103D) by incubation on a rocker for 10 min at RT. Tubes containing the bait protein immobilized on beads were placed on a magnet (Invitrogen, 12321D) and the supernatant containing excess bait protein was collected for Western blot validation. Beads were washed four times with 1x binding/wash buffer to remove any remaining excess bait protein. 5 µg of prey proteins purified human C1q (MyBioSource, MBS147305) and recombinant human p32 (LSBio, LS-G3375-20) were prepared in a 1x pulldown buffer (3.25 mM sodium phosphate, 70 mM NaCl, 0.01% Tween-20) and incubated with the immobilized bait protein on a rocker for 15 min at RT. Prior to pulldown protein capture, tubes containing the immobilized bait-prey protein complexes were placed on a magnet and washed four times with 1x binding/wash buffer to remove any remaining unbound prey proteins. Bait protein and bound prey proteins were eluted using his-elution buffer (300 mM imidazole, 50 mM sodium phosphate, 300 mM NaCl, 0.005% Tween-20) and the eluted samples were analyzed using Western blotting. The original blots are in the Source data.”

6. Figure 1E: something is not right here with the last 2 conditions: why does the isotype control IgG antibody further amplify the reducing effect of C1q on NSC proliferation and also has such a strong inhibitory effect without C1q addition?

We thank the reviewer for pointing this out. The data in this figure (Fig.1f in the revised manuscript) show that treatment of hNSC with C1q 200nM alone (grey bar) is not different from treatment of hNSC with C1q 200nM + IgG isotype control (the color of this bar has been changed from blue to grey for clarity). However, the Reviewer’s comment led us to identify an error on this graph which resulted in some confusion; we greatly appreciate the opportunity to correct this issue. Specifically a single replicate experiment in which cells were treated with isotype control IgG antibody alone (no C1q) was accidentally included; critically, not only was this data from a single well, but tracking this data back in the experimental notes identified that the chamber slide well had dried out during culture and should have been excluded altogether. We apologize for this error, and have rechecked all of the data points for this figure to ensure no additional errors are present. We have corrected Fig. 1f by removing this bar from the

graph. Again, we would highlight that there is no significant difference between hNSC treated with C1q 200nM vs. C1q 200nM + Isotype control IgG antibody groups, demonstrating that the isotype control IgG does not amplify the C1q-driven decline in proliferation. This is the relevant experiment, because fetal brain-derived hNSC do not express C1q or downstream complement pathway components, thus the addition of exogenous C1q is necessary to assess any potential effect of the isotype control.

7. Does ERK inhibition have similar effects on NSC proliferation?

In our initial report on the identification of multiple novel C1q interacting receptors, we showed that the GPCR inhibitor pertussin toxin (PTX) blocked proliferation decline but not migration (PMID: 32894219), supporting our hypothesis that C1q-mediated decline in hNSC proliferation is mediated by cell surface GPCR/s such as BAI1. However, to the reviewer's question, we also reported that Erk inhibition using a selective chemical inhibitor (PD98059) attenuated C1q [200 nM] driven changes in both migration and proliferation (PMID: 32894219), suggesting that the Erk1/2 pathway is involved in multiple cellular functions in hNSC (PMID: 12892714, PMID: 24396730).

One would predict that C1q would engage multiple of the C1q receptors identified, simultaneously activating multiple intracellular pathways depending on the relative binding affinity of each receptor for C1q and therefore C1q concentration. Additionally, three of these receptors are GPCRs, and GPCRs can also engage diverse signaling molecules, thereby modulating not only the canonical cellular responses but also noncanonical responses typically associated with activation of other cascades such as MAPK/ERK signaling (PMID: 32576977, PMID: 24396730, PMID: 29699693). In alignment with this anticipated complexity, we stated the following under the discussion: "Importantly, BAI1 may have other ligands that can also trigger quiescence in hNSC, and given that C1q internalization was only partially inhibited in BAI1 KO, other candidate C1q receptors could play a role in C1q internalization (PMID: 11827788, PMID: 28827538, PMID: 36330337)". To address the reviewer's question, we propose to expand this element of the discussion to state:

Page 8, lines 38-42, and Page 9, lines 1-3: "Importantly, while we have previously shown that GPCR inhibition via pertussin toxin (PTX) blocked C1q-induced proliferation decline but not C1q-induced migration, Erk inhibition via PD98059 modulated both migration and proliferation. These data suggest that cell surface GPCR/s such as BAI1 play a key role in C1q proliferation signaling, but also that the Erk1/2 pathway is involved in multiple cellular functions in hNSC (PMID: 12892714, PMID: 24396730). In addition, BAI1 may have other ligands that can also trigger quiescence in hNSC, and given that C1q internalization was only partially inhibited in BAI1 KO, other candidate C1q receptors could play a role in C1q internalization and modulation of cell proliferation".

8. Figure 4K: the negative C3aR/C1q control looks positive (?) or has a very high background which makes it difficult to interpret the other data – thoughts?

Thank you very much for pointing this out. It is clear that we did not select the most representative example. To allay any concerns in this regard, we have included a selection of multiple **raw** image stream examples from both C1q-C3aR negative controls and from C1q-BAI1 positive condition for positive context (Response letter_Fig.4). Cell #227 in the Response letter_Fig.3 is the original control shown in Fig. 4k, but when the brightness/contrast of the full figure was adjusted, we agree that the overlay could be misinterpreted. Accordingly, we have replaced Fig. 4k with cell #425 from C1q-C3aR negative control in the Response letter_Fig.4.

In addition, we have edited the Methods, and Online Methods to include more details as follows:

Methods, Page 16: “For imaging flow cytometry, hNSC were plated on PLO-laminin-coated 75cm² or 225cm² cell culture flasks in GM. At 70-80% confluence, hNSC were treated with 200nM C1q or an equal volume of GM (non-treated control) for 5min, 15min, 30min, 1h, 2h, or 24h followed by DPBS-/- wash and detachment using Trypsin/EDTA or non-enzymatic cell dissociation solution (Sigma-Aldrich, C5914). For matched ligand-G protein-coupled receptor C3a-C3aR complex internalization analysis, hNSC were treated with purified human C3a anaphylatoxin (Complement Technology, A118) at 100nM concentration or an equal volume of GM (non-treated control) for 5min, 15min, or 30min. For matched ligand-tyrosine kinase receptor EGF-EGFR complex internalization analysis, hNSC were treated with pHrodo™ red Epidermal Growth Factor (EGF) Conjugate (Thermo Fisher Scientific, P35374) at 40ng/mL concentration or an equal volume of GM (non-treated control) for

Response letter_Fig.4: Multiple **raw** image stream examples from both C1q-C3aR negative controls and from C1q-BAI1 positive condition.

3min. For testing C1q colocalization in endosomes or Golgi, hNSC were transfected with either CellLight™ Early Endosomes-RFP, BacMam 2.0 (Invitrogen, C10587), or CellLight™ Golgi-RFP, BacMam 2.0 (Invitrogen, C10593). The following morning, the cells were washed and treated with 200nM C1q for 24h. Cells were aliquoted and fixed using 2% PFA for 15min on ice followed by wash with 1% fish gelatin in 0.1M TBS, blocking, and 1min permeabilization using DPBS supplemented with 0.1% or 0.02% Triton X-100, 5% goat or donkey serum. Primary antibodies diluted in 0.1M TBS wash buffer with 1% fish gelatin and 5% serum were incubated with cells overnight at +4°C. Primary antibodies used were anti-C1q (1:50, Abcam, Ab71940), anti-human BAI1 (1:50, R&D systems, AF4969), anti-BAI1 (1:50, Lifespan Biosciences, LS-C120632), anti-human complement C3a (1:50, Millipore/Chemicon, CBL191), anti-C3aR (1:50, Abcam, Ab103629), anti-EGFR (Abcam, EP38y, ab52894), recombinant anti-GC1qR (1:50, Abcam, Ab24733) followed by 30min incubation RT with 1:500 dilution of secondary antibodies conjugated with 555, 488, or 647 fluorochromes (Invitrogen, A131570, A312572, A21202, A21206, A21141, A31571, A31573, A21240 or A21447). The BAI1 antibodies used for Western blotting, PLA, or imaging flow cytometry all recognize the full length BAI1 when evaluated using BAI1 over expression lysate and empty vector negative HEK293T control lysate (Novus Biologicals; Supplementary Fig.3a). Single cells at a density of 20,000 cells/μl were imaged using Amnis Imagestream x Mark II Imaging Flow Cytometer and analyzed using IDEAS version 6.2 according to the manufacturer instructions (Cytex Biosciences)¹⁰⁸. Gating were set using unstained negative control cells, secondary antibody only controls, single labeled cells, and antibody compensation beads (Life technologies, A10513). For a gating examples see also Supplementary Fig.3. For protein internalization, score ≥ 0.5 was considered internalized. Time course data was collected from one to four independent experiments with one sample per each. Each independent experiment consists of one flask treated for independent times or plated and treated on independent days.”

“Online Methods: Page 30, Ligand/receptor internalization analysis using imaging flow cytometry

Single cells at a density of 20,000 cells/μl in 0.1M TBS wash buffer with 1% fish gelatin were imaged using Amnis Imagestream x Mark II Imaging Flow Cytometer and analyzed using IDEAS software version 6.2 according to the manufacturer instructions (Cytex Biosciences)¹⁰². Briefly, for ligand/receptor internalization analysis cells were eroded using an eroding mask (Fig.4e) of 7-pixel (Fig.4b,c,o-q, and Supplementary Fig.4b,c,f,g, o) or 3-pixel on a brightfield channel (Fig.4d,f and Fig.4u, and Fig.5 a,b). Cells with a ligand/receptor internalization score ≥ 0.5 was gated as internalized (Supplementary Fig.3b, Supplementary Fig. 4c,g,o). For colocalization analysis, cell population exhibiting ligand and receptor

internalization were plotted on a histogram using bright detail similarity R3 feature designed to specifically to compare the small bright image detail of two images. This feature is the log transformed Pearson's correlation coefficient of the localized bright spots with a radius of 3 pixels or less within the masked area in the two input images. Since the bright spots in the two images are either correlated (in the same spatial location) or uncorrelated (in different spatial locations), the correlation coefficient varies between 0 (uncorrelated) and 1 (perfect correlation) and does not assume negative values. The coefficient is log transformed to increase the dynamic range between (0, infinity). In this study, mean bright detail similarity R3 score (co-localization score) ≥ 1.5 was considered colocalized (Fig.4d,o,u, and Supplementary Figs.3c and 4d,h). Spot count feature (Supplementary Fig.4o) was used for measuring the quantity of internalized protein per cell (Fig.4q, Fig.5b, Supplementary Fig.3c)."

"Fig.4: Endocytic internalization of C1q-BAI1 complex allows intracellular C1q-p32 interaction and p32 instability.

a-c,e, Timecourse Imagestream analysis of C1q internalization kinetics in immunolabeled C1q [200 nM]-treated and non-treated control hNSC (a-c) using 7-pixel component masking visualized in e ($n = 2-5$ independent experiments per timepoint). Cells with a ligand internalization score ≥ 0.5 were gated as internalized. **d,f-g**, Co-localization of intracellular C1q immunolabel and Rab5-RFP reporter transfected hNSC endosomes (d,f) vs. Golgi-RFP (g) with 3-pixel component masking. Co-localization score (bright detail similarity R3 score) ≥ 1.5 was considered colocalized....**j-k**, Intracellular ligand-receptor co-localization after immunolabeling of matched ligand-receptor positive control C3a-C3aR (**j**) vs. mis-matched negative control C1q-C3aR (**k**). **l,n-o**, Timecourse analysis of C1q-BAI1 complex internalization kinetics in BAI1⁺ in-focus single cells using component masking ($n = 2-3$ independent experiments per timepoint). **p-q**, BAI1 KO decreases the proportion of in-focus single hNSC with intracellular C1q (**p**) and the overall quantity of internalized C1q per cell within this sub-population analyzed (**q**) relative to C1q [200 nM]-treated BAI1 WT (ctrl, dashed line) ($n = 3$ independent experiments)."

"Supplementary Fig.4: C1q-bait-intracellular prey-protein pull-down data and Imagestream analysis validation. Related to Fig.4 and 5. **b-e**, Timecourse analysis of matched ligand-G protein-coupled receptor C3a-C3aR complex internalization kinetics in C3a⁺ in-focus single cells post-C3a treatment. **f-j**, Matched ligand-tyrosine kinase receptor EGF-EGFR complex internalization in EGF⁺ in-focus single cells post-phRodo EGF-treatment in hNSC. An Imagestream example of pHrodo-EGF and EGFR ICC staining (**j**). For ligand/receptor internalization analysis cells were eroded using an eroding mask of 7-pixel on a brightfield channel and cells with a ligand/receptor internalization score ≥ 0.5 were gated as internalized. **k-l**, Timecourse Imagestream images of C1q-BAI1 complex

internalization kinetics in BAI1+ in-focus single cells 15min (**k**) and 1h post-C1q treatment in BAI1 WT hNSC (**l**). m-n, C1q-BAI1 complex internalization kinetics in-focus single cells 1h post-C1q treatment in BAI1 KO hNSC (**m**), and example images of negative control background in non-treated, C1q-stained hNSC vs. C1q-treated, 2. ab only-stained hNSC (**n**). **o**, An example of Imagestream component masking for spot counts in the intracellular compartment to compare intracellular C1q quantity. **p**, Imagestream analysis of p32 cellular localization and mean internalization score in non-stained vs. p32 stained hNSC based on IDEAS software.”

9. Figure 7: the analysis of C1q is very nice – why did the authors not perform the same analysis for BAI1? It would be nice to see the expression profile of this C1q receptor in relation to its ligand. It would have also added some suggestions to whether the interaction by be autocrine or paracrine.

This is an interesting idea, unfortunately, in our hands, transcriptomic analysis is a poor way to look at the expression of most of the C1q receptors we have identified. This is particularly true for the GPCRs, which are highly stable proteins in the cell membrane; as a result, transcription is maintained a level that is low or even undetectable in RNAseq.

However, as per the Reviewer’s suggestion, we analyzed local Bai1 mRNA expression in acute SCI by different cell types using published scRNAseq data (PMID: 34132743). scRNAseq shows low Bai1 reads mostly detectable in spinal cord neural cell populations, such as astrocytes, OPC/oligodendrocytes, ependymal, and neurons, and some immune cells (R_Fig. 4A), and suggests that SCI does not have a large impact on Bai1 expression (R_Fig. 4C). In contrast, C1qa in this scRNAseq dataset is highly expressed by immune cells in both intact and injured spinal cord (R_Fig. 4C), and nearly all immune cell populations and spinal cord neural cell types upregulate C1qa after SCI (R_Fig. 4D). To the Reviewer’s point, these data suggest a predominantly paracrine signaling mechanism for C1q-BAI1 interaction in spinal cord cells during homeostasis and after injury (1dpi). However, subacute upregulation of C1qa reads post-SCI (3-7dpi) may also allow C1q-BAI1 autocrine interactions.

We are currently at max capacity for Supplementary Figures, however, if the Editor allows, we are more than happy to include the Response letter_Fig. 5 and this discussion into our MS as a Supplementary Figure.

Response letter_Fig. 5 C1qa-Bai1 expression in spinal cord cell types in intact or spinal cord injured mouse. **a-b**, UMAP plot of expression of Bai1 vs. C1qa in major spinal cord cell populations based on scRNAseq (PMID: 34132743). Low reads of Bai1 are mainly detectable in spinal cord neural cell populations, such as astrocytes, OPC/oligodendrocytes, ependymal, and neurons, and some immune cells (**a**). C1qa is highly expressed by immune cells in both intact and injured spinal cord (**b**). **c-d**, Dot plot of abundance of Bai1 (**c**) or C1qa (**d**) transcripts per cell types in intact or injured spinal cord (PMID: 34132743). Each dot size represents percentage of the detected transcripts, while the color intensity indicates log-normalized counts per cell type. SCI does not have a dramatic impact on Bai1 expression (**c**). Nearly all immune cell populations and neural cell types upregulate C1qa after SCI (**d**). These data highlight the likelihood of paracrine signaling for C1q-BAI1 interaction during homeostasis and acutely 1dpi, however, subacute upregulation of C1qa reads post-SCI may also allow C1q-BAI1 autocrine interactions.

10. *The in vivo experiment in Figure 7 solidifies the role of BAL1 in hNSC proliferation control. It does not conclusively allow to associate this with the C1q activity though. Could the authors approach this somehow? For example, could there be a bulk RNA-seq analysis of in vitro WT and BAL1 KO hNST +/- C1q (as in Figure 6) to generate C1q-BAL1 axis (normal/perturbation) signatures and then utilize these signatures to assess the bulk RNA-seq signatures from re-isolated SC121-BrdU+ cells WT/KO cells in the lesions (Figure 7D)?*

We appreciate the reviewers thoughts on this matter, as we have found this issue challenging to approach. We have previously attempted to re-isolate engrafted hNSC (PMID: 28479305). However, because these cells begin to migrate and differentiate quickly after transplantation (PMID: 24936450), the yield even at 2 weeks post-transplantation is incredibly low. For example, 900 cells pooled from 2-5 transplanted animals, out of 50,000 cells total transplanted (PMID: 28479305). We feel therefore that this approach would not give an unbiased or reproducible sample from which to conduct a comparison. While we absolutely agree in principle that an ideal experiment would definitely tie in vitro definition with mechanism to in vivo action, we do not see a path to overcome this hurdle. For this reason, we have also a section in the discussion noting that there may be other BAL1 ligands present *in vivo* that modulate the mechanism we describe in vitro.

11. *The manuscript has many errors when it comes to labeling and cross-referencing figures and data. For example, the table is references erroneously as Extended Data 2, there is no reference to Extended Data 3 in the text and Extended Data 8 (mentioned in the text) are entirely missing, Extended Data 7 legend seems not to fit (ROS), etc.. Overall, the structure of text, figures, and Extended data can be improved – it was at times difficult to navigate text and figures.*

We apologize, and thank the reviewer sincerely for highlighting for us so that we could address these errors; We have corrected the numbering of the Supplementary Figures.

Reviewer #3 (Remarks to the Author):

In this work the authors focus on the complement factor C1q and its effects on proliferation and quiescence of human neural stem cells (hNSCs). First, they demonstrate that C1q treatment leads to these cells proliferating significantly less with the absence of any changes in apoptosis, suggesting an induction of quiescence. Next, they undertake a variety of studies to assess the underlying mechanism of this effect. The first of these mechanisms is shown through changes in p53 phosphorylation and localization to the nucleus which seems to occur through MDM2 initiated by binding to BAI1. They then assess the internalization of C1q and suggest that this occurs via endocytosis. The authors also detail a novel role for C1q in mitochondrial changes in hNSCs acting through p32 and BAI1. Finally, the authors show that loss of BAI1 rescues the C1q impairments in proliferation and that transplantation of BAI1 KO cells are more highly proliferative in a spinal cord injury model. These data are exciting and

expand our understanding of the role of immune molecules in the control of proliferation and quiescence of neural stem cell populations. While I am enthusiastic about these studies, I do have a number of criticisms that the authors should consider.

-In Figure 1E, it is shown that IgG nab both alone or in the presence of C1q leads to a ~5-fold larger reduction in BrdU positive cells than the when C1q is used alone. How can this be as this is a control antibody?

We thank the reviewer for pointing this out. The data in this figure (Fig.1f in the revised manuscript) show that treatment of hNSC with C1q 200nM alone (grey bar) is not different from treatment of hNSC with C1q 200nM + IgG isotope control (the color of this bar has been changed from blue to grey for clarity). However, the Reviewer's comment led us to identify an error on this graph which resulted in some confusion; we greatly appreciate the opportunity to correct this issue. Specifically a single replicate experiment in which cells were treated with isotype control IgG antibody alone (no C1q) was accidentally included; critically, not only was this data from a single well, but tracking this data back in the experimental notes identified that the chamber slide well had dried out during culture and should have been excluded altogether. We apologize for this error, and have rechecked all of the data points for this figure to ensure no additional errors are present. We have corrected Fig. 1f by removing this bar from the graph. Again, we would highlight that there is no significant difference between hNSC treated with C1q 200nM vs. C1q 200nM + Isotype control IgG antibody groups, demonstrating that the isotype control IgG does not amplify the C1q-driven decline in proliferation. This is the relevant experiment, because fetal brain-derived hNSC do not express C1q or downstream complement pathway components, thus the addition of exogenous C1q is necessary to assess any potential effect of the isotope control.

...Moreover, in this same figure it's shown that after priming for 4-8 weeks this actually leads to greater proliferation (ie. BrdU+ cells) than control. Can the authors explain why this might be? Are the spheres just much larger but there are many fold less of them?

We thank the Reviewer for their interest in division modes of NSC driving self-renewal vs. differentiation. We agree that the opposite effects of long-term C1q priming on proliferation vs self-renewal are intriguing.

Different modes of division drive self-renewal vs. differentiation in NSC - symmetric cell divisions produce identical progeny, while asymmetric divisions produce two different types of progeny (PMID: 35693936). NSC division can therefore be any of the following: self-renewing symmetric, which produces two new NSC; self-renewing asymmetric, in which one of the two cells remains as an NSC and another initiates terminal differentiation; or self-consuming symmetric, which produces two faster proliferating transit amplifying (TA) progenitors that both initiate terminal differentiation. Self-consuming symmetric cell divisions leading to terminal differentiation can thus deplete the originating stem cell population and result in stem cell exhaustion. Therefore, our observations in which long-term C1q priming causes a proliferation rebound and increase coupled with an overall decline in self-renewal could be interpreted as NSC

shifting their mode of division from self-renewing to self-consuming. This would suggest that extended exposure to C1q at blood plasma concentrations could affect maintenance of the NSC pool and long-term tissue turnover, providing an important insight into how C1q could contribute to CNS pathological processes and impact repair capacity. Critically, decline of the neural stem cell pool is posited to be a major factor in age-related decline of regenerative functions in the CNS, and is thought to derive from increased quiescence and terminal differentiation (PMID: 31805262, PMID: 36922629). We have a follow-up manuscript in preparation demonstrating importance of C1q in stem cell self-renewal, proliferation, and neuronal maturation in aging brain.

While we alluded to this in the original manuscript, we recognize that the description was quite brief. Accordingly, we have expanded our interpretation of C1q long-term priming effect on NSC proliferation rebound vs. decline in self-renewal in the Results to better reflect this context in Page 3, lines 9-23.

As per the Reviewer's suggestion, we have also added neurosphere size to Supplementary Fig.1f,g. We did not detect a significant change in neurosphere size during C1q [200nM]-treatment relative to non-treated controls (Response letter_Fig.6a). Similarly, long-term C1q [200nM] priming caused no significant differences in neurosphere size relative to non-treated controls - with the exception of the observation that by 6 weeks of C1q exposure no neurospheres were detected, resulting in a size score of virtually 0 (Response letter_Fig.6b).

Response letter_Fig.6. a, Neurosphere size in C1q-[200nM] treated hNSC relative to non-treated controls ($n = 3$ independent experiments). b, C1q-[200nM] priming effect on neurosphere size relative to non-treated controls after medium switch and analysis under normal growth conditions ($n = 4$ independent experiments). In absence of neurospheres, the sphere area for analysis was scored virtually 0. Mean \pm s.e.m., a 1-sample 2-

tailed t-test, b Kruskal-Wallis test with post hoc and 1-sample 2-tailed t-test. ns = not significant, $p > 0.05$; $*p \leq 0.05$. See also the Source data.

These data support the interpretation that there are fewer sphere initiating cells present over time in this clonal assay, and are generally consistent with a potential shift to self-consuming symmetric divisions. A description of these added data has been included under the results together with the expanded interpretation of C1q long-term priming effect on NSC proliferation rebound vs. decline in self-renewal (from Page 2 lines 29 to Page 3, line 6).

“A reversible loss of proliferation combined with cell cycle arrest at G0 in the absence of cell death (Fig.1a-j) is consistent with stem cell quiescence^{24,25}, which would predict a decrease in self-renewal. As hypothesized, clonal neurosphere assays revealed a significant decrease hNSC self-renewal during C1q [200nM] exposure vs. non-treated controls (Fig.1k,l). No significant change was detected in neurosphere sizes during C1q [200nM]-treatment relative to non-treated controls (Supplementary Fig.1f). Additionally, stem cell quiescence is distinct from cell cycle arrest during terminal differentiation. Analysis of total protein lysates from hNSC exposed to C1q [200nM] during in vitro differentiation revealed significant increases in undifferentiated NSC/progenitor markers, nestin and GFAP δ 26, but not lineage commitment markers (Fig.1m). RNAseq further identified transcriptomic enrichment of quiescence genes in C1q [200nM] treated hNSC (Fig.1n, grey). These data indicate that C1q induces quiescence in hNSC.

While the effect of C1q on NSC proliferation was reversible after short-term exposure (up to 48h, Fig.1e), CNS injury/aging can cause prolonged C1q elevation. We therefore tested whether long-term exposure to C1q has a priming effect on hNSC proliferation or self-renewal. hNSC neurosphere cultures were exposed to C1q for 2, 4, 6, or 8 weeks, at which time C1q was withdrawn and cells returned to normal in vitro growth conditions for analysis 5 weeks later. Consistent with short-term exposure effects (Fig.1e), hNSC proliferation declines were reversed after C1q withdrawal (Fig.1o). hNSC self-renewal declines (Fig.1l), however, persisted after long-term priming (Fig.1p), and were exacerbated by length of exposure. Long-term C1q [200nM] priming caused no significant differences in neurosphere size relative to non-treated controls (Supplementary Fig.1g) supporting the interpretation that there were fewer neurosphere initiating cells present in C1q [200nM] primed group over time.”

In order to further clarify, we have also expanded the experimental methods and figure legend, in order to make sure that the paradigm for Fig.1o,p in the revised manuscript is clear. For Fig.1o, hNSC were cultured as neurospheres through treatment with C1q, the C1q washed out, and then cells were plated in monolayer for BrdU analysis of proliferation. For Fig.1p, cells received the same treatment with C1q, but after

dissociation to single cells, were placed as single cells on low adherence plated to analyze neurosphere initiating capacity.

We have also edited the Methods and Figure legends for clarity as follows:

Methods: Proliferation, neurosphere formation, and RealTime-Glo Annexin V multiplex assay “For timecourse C1q priming assays, hNSC were co-cultured as free-floating neurospheres with or without purified C1q 48h or for up to 6-8 weeks followed by C1q wash out and single cell plating in monolayer in GM at a density of 15,000 cells/well for BrdU or low adherence in GM at a density of 15 cells/well for neurosphere initiating capacity. Post-C1q wash BrdU incorporation time under normal growth conditions was for 48h.”

“Fig.1**o-p** Timecourse C1q-priming experiments followed by a medium switch and analysis under normal growth conditions show long-term effect of C1q on hNSC proliferation in monolayer culture (**o**) and self-renewal in free-floating neurospheres (**p**) vs. controls ($n = 3$ or 4 independent experiments).”

-In a number of cases, western blots are shown with bands that come from a number of different blots including when trying to demonstrate quantitative differences (for example Figure 1K). In such cases, these data need to come from one blot.

Quantitive comparisons and example western blot bands shown in this manuscript are from the same blots as shown in the Source data. Experimental conditions that are not part of this study have been cropped out from the example blots in Fig.1m, Fig.2b, Fig.4y, Fig.5f, and Supplementary Fig.3a in the revised manuscript. We have revised the figures to indicate this clearly by adding grey line between the lanes. Revised figure legends also include this detail: “Experimental conditions that are not part of this study have been cropped out from the example blots (grey line between the lanes) or Experimental conditions have been cropped out from the example blots for better visualization (grey line between the lanes). The full original blots are in the Source data.”

-In multiple figures (for example, Figure 2c and Figure 3d) no control images are shown and only treatment examples are used. This makes interpretation of the data difficult. In the case of Figure 2c, there is an apparent 4-fold increase in the presence of C1q which should be striking in images comparing Control to 200 nM C1q.

We understand the reviewer’s point completely, but the limited space available makes this a challenging issue to address. The intent of showing fluorescent labeling for p53ser15 under Fig.2c, for example, was simply to show that nuclear localization was quantifiable, as shown in the associated graph. To make that clearer, we have revised this Fig.2 to label 2c separately from the graph. Accordingly, the figure legend now states:

“Fig. 2: Blood C1q concentration activates p53 signal transduction driving cell cycle quiescence. **c**, An example image of phospho-p53Ser15 nuclear localization in hNSC after ICC. **d**, Quantification of ICC for phospho-p53Ser15 nuclear localization in hNSC treated with C1q [200nM] or T^oInact C1q [200nM] (negative control) normalized to non-treated controls (n = 3-5 independent experiments). Mean \pm s.e.m., **d** Welch’s ANOVA with post hoc test and 1-sample 2-tailed t-test.

We have revised phospho-p53Ser15 ICC figure legend also in Fig. 3:

“Fig 3: C1q-driven p53 signal transduction is BAI1-dependent. **j**, Example images of phospho-p53Ser15 staining in Dapi+ nucleus vs. cytoplasm with Mitotracker+ mitochondria. **k**, Quantification of phospho-p53Ser15 localization in nucleus in C1q-treated BAI1 WT or KO hNSC using ICC relative to each control (n = 3-4 independent experiments). Mean \pm s.e.m, **k** 1-way ANOVA with post hoc test.

We would be happy to add treated and non-treated images to the extended data if this is judged to be a critical issue.

In Fig.3d PLA data, we have a similar situation. There is insufficient space to be able to include the 4 conditions quantified in panel Fig.3c. Accordingly, because the treated condition has the most potential for background, we showed only treated WT versus KO, which we felt would provide a conservative and realistic view by which to judge the quantified results. Because there are no quantifiable differences detected between control WT versus BAI1 KO, or control versus treated BAI1 KO, we feel that these images are sufficient to support the quality of the staining - which was the intended goal. To make that clearer, we have revised Figure 3 legend:

“Fig.3: C1q-driven p53 signal transduction is BAI1-dependent.

c-h, PLA analysis of protein interaction between C1q-BAI1 (**c**), BAI1-MDM2 (**e**) or MET-MDM2 negative control (**h**) in C1q-treated vs. control BAI1 WT or KO hNSC. Example images showing PLA signal (red dots) for C1q-BAI1 (**d**), BAI1-MDM2 (**f**), or MET-MDM2 (**g**) in C1q-treated BAI1 WT vs. KO hNSC (n = 3-4 independent experiments). **c** 1-way ANOVA with post hoc test, **e** Welch’s ANOVA with post hoc test, **h** Kruskal-Wallis test.

Again, we would be happy to add treated and non-treated images to the extended data if this is judged to be a critical issue.

-While the BAI1 KO and C1q dependence in this work is convincing the lack of a direct demonstration of C1q and BAI1 interaction is a significant weakness. Considering that BAI1 recombinant protein is commercially available it should be straightforward to demonstrate a direct interaction using a technique such as biolayer interferometry or surface plasmon resonance or a more traditional pull-down assay. It remains possible that the interaction between BAI1 and C1q is indirect.

We thank the reviewer for this thoughtful comment. We agree that a more detailed analysis of the interaction kinetics between C1q and BAI1 is important. To address the reviewer's immediate concerns, we performed an *in situ* pull-down using recombinant human C1q and BAI1 to demonstrate direct C1q-BAI1 interaction (Response letter_Fig.7). These data demonstrate direct interactions between C1q-BAI1, C1q-p32, and BAI1-p32, and that C1q-BAI1-p32 can be pulled down as a complex. The data is included in the revised manuscript as Fig.5f. We have a strong interest in studying C1q interaction kinetics with all of the novel receptors we have identified (PMID: 32894219) using either bio-layer interferometry (BLI) or Surface Plasmon Resonance (SPR) ; however, the expense of commercially available C1q and amount of protein (\$8000/10mg) necessary requires establishing affinity column chromatography purification of C1q from human plasma in our lab at UCI in order to complete those studies. While in progress, we did not want to delay our resubmission further given the large amount of data we have already incorporated to the revision.

Response letter_Fig. 7. Direct interaction between C1q-BAI1, C1q-p32, and BAI1-p32 tested using *in situ* pull-down of human purified or recombinant proteins.

-The model with which p32 is involved with C1q and BAI1 doesn't seem fully plausible. The author's own data suggests that p32 is in the mitochondrial matrix while C1q in large part enters the cell via endosomes. How can these two proteins specifically interact when they are in different cellular compartments? This would require demonstrating that both C1q are in the same place at the same time in hNSCs at high resolution. Moreover, if the authors propose, as in Figure 5e, that C1q enters endosomes and then is released from endosomes this would need to be demonstrated experimentally.

In this manuscript we focus on whether internalized C1q interacts with intracellular p32 in a BAI1-dependent manner. While we show that C1q is initially internalized via Rab5+

endosomes, a specific intracellular localization for C1q-p32 interaction remains to be determined. However, we thank the reviewer for this insightful question, as intracellular protein interactions and trafficking are areas where much of the field of cell biology is seeking new understanding.

In endocytosis, the material to be internalized from the extracellular region is surrounded by an area of plasma membrane, which then buds off inside the cell to form a vesicle/vacuole containing the ingested material. The vesicle trafficking pathway includes not only vesicle formation but also vesicle transport along the cytoskeletal track, and vesicle fusion with the target membrane. Endocytic membrane trafficking and signaling are interconnected, and cargo sorting/signaling can take place at multiple platforms from early/recycling endosomes to late endosomes / MVBs (PMID: 36129576). Endosomes communicate with other organelles through membrane contact sites (MCS). Most notably, tubules emanating from the endoplasmic reticulum (ER) contact endosomes within a distance of 30 nm or shorter (PMID: 32258039). In this, the ER is a dynamic organelle that performs many functions including, lipid metabolism, calcium storage, and protein folding - and also makes contact sites with multiple membranes including mitochondria (PMID: 26041457).

So far, we know that p32 protein is translated in the ER as a precursor (pre-p32) containing a mitochondrial targeting sequence which is proteolytically removed (PMID: 9305894, PMID: 9531316). Even though, p32 is mainly localized in the mitochondria, it has been also detected in the ER (PMID: 11493647), nucleus (PMID: 11243856), and cell surface. Due to the localization of p32 in these different subcellular compartments, and the capacity of p32 to interact with several proteins, p32 is thought to function as a chaperone protein (<https://www.uniprot.org/uniprotkb/Q07021/entry#interaction>). In agreement with this interpretation, we report detection of p32 in both mitochondrial and insoluble/particulate fraction of subcellular structure bound proteins but not at the cell surface (Fig.4r,s).

In hNSC, internalized C1q is detected at least in endosomes and endosomal multivesicular bodies (Fig.4f,h). We completed unbiased nanoLC-MS/MS of pull-down experiments in hNSC using internalized C1q as bait, seeking to identify intracellular interacting proteins (prey). We then conducted subcellular enrichment analysis of prey proteins that were significantly different in C1q vs untreated samples, identifying several specific compartments involved in receptor-mediated endocytic trafficking of C1q (Fig. 4i, Supplementary Fig.4a). These data suggest that internalized C1q is present in several locations permitting interaction with p32. However, there are many additional questions for future studies, including identification of target membranes, tethering protein/protein complexes coordinating C1q cargo, specific locations of C1q-p32 interaction, and the biological processes regulated by these interactions.

Minor:

-Page 2 Line 1 BCB not defined. Perhaps BSB typo?

We appreciate this comment and have corrected blood-central nervous system barrier (BCB) to blood-brain/spinal cord-barrier (BBB/BSB) in page 2 line 2, and check the rest of the manuscript for consistency regarding the terminology.

-In a number of cases “Blood-derived C1q concentrations” is used. This is confusing as the data shown here actually has nothing to do with “Blood-derived C1q concentration promotes.” Consider modifying this terminology for clarity.

Per Reviewer’s request, we have changed the original subtitle “Blood-derived C1q promotes hNSC quiescence” to “C1q promotes hNSC quiescence” page 2 line 11.

-In Figure 3B, this is the only blot shown as white bands on black background when all of the rest of the blots are shown inverted. This is confusing and can lead to confusion over whether these data are western or an agarose gel.

Agreed; We have changed Fig. 3b to match the style of rest of the western blots.

-In Figure 4i, pull down based proteomics data is introduced but these data are hardly described or mentioned at all and only shown as a cartoon. If these data are significant enough to be shown in the main figures they should be described and interpreted in a more fulsome manner.

We thank the Reviewer for bringing this to our attention - we recognize that the description of Fig.4i was quite brief. To address this comment, we have changed the figure format into a bubble plot for clarity and expanded the text accordingly. Briefly, these data represent unbiased nanoLC-MS/MS of pull-down experiments in hNSC using internalized C1q as bait, seeking to identify intracellular interacting proteins (prey). We then conducted subcellular enrichment analysis of prey proteins that were significantly different in C1q vs. untreated samples, identifying several specific compartments involved in receptor-mediated endocytic trafficking of C1q (Fig.4i, Supplementary Fig. 4a; Response letter_Fig.8). The complete list of proteins that were significantly different in C1q vs. untreated samples, and the subcellular compartments identified using subcellular enrichment analysis, is now shown as Supplementary Fig.4a and is also included in the source data.

Response letter_Fig.8. A bubble plot showing subcellular enrichment of prey proteins that were significantly different in C1q vs. untreated samples in several specific compartments involved in receptor-mediated endocytic trafficking of C1q in hNSC after unbiased nanoLC-MS/MS analysis. A FDR-adjusted p-value threshold of ≤ 0.05 was considered significant (n = 4 or 8 independent experiments).

In order to further clarify, we have also expanded the figure legend as shown in Response letter_Fig 8., and experimental methods, in order to make sure that the paradigm for Fig.4i in the revised manuscript is clear.

Methods, Page 17: "Cross-linking, co-immunoprecipitation, and LC-MS/MS

500 μ g of C1q was reversibly biotinylated with an Ez-link NHS-SS-Biotinylation kit (ThermoScientific, PI21441) following the manufacturer instructions. Briefly, to prepare biotin-labeled C1q, 1mg purified human C1q was resuspended in 1mL Milli-Q H₂O (1mg/mL) and half of the vial (500 μ g of C1q) was buffer exchanged to PBS using Zeba spin desalting column 7K MWCO (ThermoFisher Scientific) following the manufacturer instructions. C1q in PBS (1.25nM C1q) was incubated with a 20-fold molar excess (25nM) of NHS-SS-Biotin for 1h RT in agitation. The non-bound biotin was removed by buffer exchange using a Zeba spin desalting column 7K MWCO (500 μ l final volume). 10 μ l aliquot (1:30 dilution) was used to determine C1q protein concentration by BCA analysis. A total of 60-80% of the original amount of C1q was recovered after the biotinylation process. An aliquot of the biotinylated C1q was analyzed by Western blot to identify biotinylated C1q.

hNSC were grown as a monolayer at PLO-laminin-coated 6-well plates. Prior to the experiment, the cells were washed three times with PBS (pH 7.2) and 200nM of biotinylated C1q (bait) or an equal volume of GM (non-treated controls) was added to the cells for 15min or 30min at RT. Analysis was performed as four independent experiments with one sample well per each. A cell membrane permeable DSP with 12Å spacer arm (ThermoScientific, PI22586) was used for intracellular crosslinking of the treated samples according to the manufacturer instructions. Briefly, the cells were washed three times with PBS (pH 7.2) and exposed to 2mM of DSP at RT for 30min. The sample was quenched with three 5min washes of 25mM Tris in PBS. The cells were lysed using Pierce IP lysis buffer (ThermoScientific, PI87787) supplemented with protease and phosphatase inhibitors cocktail on ice. Lysates were centrifuged 13,000g for 10min at 4°C and for pull-down supernatants were incubated on washed Pierce Streptavidin Magnetic beads (ThermoScientific, PI188816) for 1h at RT followed by washes to remove non-specific binding partners. For proteolytic protein digestion, magnetic beads with protein were washed with HPLC grade H₂O. 2M urea with 25mM (NH₄)HCO₃ was added into slurry in 1:1 ratio followed by 15min incubation with DTT as a final concentration of 2mM at 37°C. Next, the

slurry was incubated with iodoacetamide at final concentration of 4mM for 30min protected from light followed by 15min quenching with DTT as a final concentration of 2mM at 37°C. 25 mM (NH₄)HCO₃ was added into the slurry in 1:1 ratio to dilute urea into final 1M followed by overnight incubation with trypsin at a final concentration of 20ng/μL at 37°C. Digestion was stopped by 0.1% formic acid and the protein solution was desalted using C18 column (Agilent) according to the manufacturer instructions.

Peptide digests were subjected to LC MS/MS analysis using an UltiMate 3000 RSLC system (Thermo Fisher Scientific) coupled in-line to an Orbitrap Fusion Lumos mass spectrometer (Thermo Fisher Scientific). Reverse-phase separation was performed on a 50 cm x 75 μm I.D. Acclaim® PepMap RSLC column. Peptides were eluted using a gradient of 4% to 22% B over 70min at a flow rate of 300nL/min (solvent A: 100% H₂O, 0.1% formic acid; solvent B: 100% acetonitrile, 0.1% formic acid). Each cycle consisted of one full Fourier transform scan mass spectrum (375–1500 m/z, resolution of 120,000 at m/z 400) followed by data-dependent MS/MS scans acquired in the linear ion with HCD at NCE 30% trap at top speed for 3s. Target ions already selected for MS/MS were dynamically excluded for 30s. Protein identification and label-free quantitation were carried out using MaxQuant as described¹⁰⁹. Raw spectrometric files were searched using MaxQuant version 1.6.0.16 against a FASTA of the complete human proteome obtained from SwissProt Feb 2020 version. The first search peptide tolerance was set to 20 ppm, with main search peptide tolerance set to 4.5 ppm. Trypsin was set as the digestive enzyme with max 2 missed cleavages. Methionine oxidation and protein N-terminal acetylation were set as variable modifications, while cysteine carbamidomethylation was set as a fixed modification. Peptide spectra match and protein FDRs were both set as 0.01. The generated raw mass spectrometry data was preprocessed and analyzed by Differential Enrichment Pipeline package (DEP) version 1.22 (Bioconductor)¹¹⁰. For quality control included filtering proteins with high missing values, assessing reproducibility, and addressing batch effect, the tool's default settings were applied. Quantitative values were normalized using variance stabilizing transformation (VST) to minimize technical variation and improve comparability across samples. For differential expression analysis, the samples treated with biotinylated C1q per each independent experiment were pooled together and protein-wise linear models combined with empirical Bayesian statistics, as implemented in the DEP package, were used to identify differentially expressed proteins between two conditions bait versus non-treated control. Adjusted p-values were calculated using the Benjamini-Hochberg method, with proteins showing an FDR-adjusted p-value ≤0.05 considered significantly differentially expressed. The significant proteins are listed in the Source data. Hypergeometric tests in SubCellViz R package¹¹¹ was employed to analyze the subcellular localization of the

differentially expressed proteins to illustrate protein distribution and identify compartment-specific functional patterns. Protein identifiers were annotated to subcellular compartments using Gene Ontology Cellular Component (GO-CC) and UniProt databases. Enrichment analysis was performed using hypergeometric tests, and compartments with FDR-adjusted p-values ≤ 0.05 were identified as significantly enriched. All analyses were performed by investigators blinded to experimental conditions/groups.”

Reviewer #4 (Remarks to the Author):

"I co-reviewed this manuscript with one of the reviewers who provided the listed reports. This is part of the Nature Communications initiative to facilitate training in peer review and to provide appropriate recognition for Early Career Researchers who co-review manuscripts."

Thank you, we appreciate the Reviewer's time and effort to improve our manuscript.

Authors response to Reviewers

We appreciate the valuable feedback on our manuscript. We have carefully considered each comment and made revisions accordingly. Additionally, as requested, all of the figures have been formatted to match Nature Communications requirements, and the source data files with raw data have been included in this resubmission. Below, we address each of the points raised.

REVIEWER COMMENTS

Reviewer #1 (Remarks to the Author):

This manuscript reports effects of C1q on human neural stem cells (hNSCs) mediated by the receptor BAI1. The authors describe effects of C1q on hNSC proliferation and other aspects of the cells' physiology. In parallel studies, the authors provide insight into the signaling pathways that are engaged when C1q binds BAI1, specifically i) regulation of mdm2 to control p53 and ii) endocytosis of C1q to modulate p32. These findings are novel and interesting.

The authors were very responsive to the reviewers' comments and have addressed all of the reviewers' concerns following the initial review. The textual changes and new data added by the authors have significantly strengthened the manuscript.

We thank Reviewer for their time and effort invested to improve our manuscript.

Reviewer #2 (Remarks to the Author):

The authors have addressed to the largest extent the queries and concerns raised by this reviewer satisfactorily.

The only 'negative' remaining would be that the authors have missed an opportunity pinpoint then intracellular location of the C1q/p32 location better - this was also raised by reviewer 3. I am not sure why as this would have elevated to the work substantially and performing higher resolution/quality staining of mitochondria as suggested by this reviewer (or endosomal compartments as indicated by the authors) is not difficult. However, this reviewer does not feel sufficiently strongly about this matter to ask for another revision.

We thank Reviewer for the insightful questions, and the time and effort invested to improve our manuscript. Based on our current knowledge of the possible intracellular localization of p32 and the several specific intracellular compartments involved in receptor-mediated endocytic trafficking of C1q intracellular we hypothesize that the location of C1q-p32 intracellular interactions is far more complex than anticipated (see C1q-p32 PLA interaction puncta in Fig.4V). Therefore we would prefer to address this question in a follow-up manuscript in preparation reporting additional novel biological processes regulated by intracellular C1q.

Reviewer #3 (Remarks to the Author):

The authors have suitably addressed a number of my concerns and the manuscript has improved substantially. However, a couple of points remain:

1) I congratulate the authors for carrying out the pull down experiment using each of the interacting proteins. These data are now in Figure 5F. The manner, however, in which these data are presented makes it nearly impossible to interpret. For example, it appears that BAI1 is used as the bait in all lanes (except 3 and 4) shown in these blots and the probed with antibodies above? The presentation of these data could be significantly improved by indicating bait and prey in each case. It took this reviewer, a significant amount of time reviewing the methods and looking at the figure and legend to get an idea of what is going on with that data. The data should be interpretable by simply looking at the figure and legend. In addition, in lanes 3 and 4, it appears that no BAI1 bait was used and yet C1q and p32 are pulled down. How can that be if only BAI1 has the his-tag to bind to the magnetic beads? Is this result of non-specific binding of C1q/p32 to the beads or something else? If the former, this would seriously impact these data and call them into question.

We thank Reviewer for their extra time and attention to detail to improve the readability of our manuscript. We utilized His-tagged BAI1 as the bait for all pulldowns except the C1q-p32 pulldown. To test C1q-p32 interactions, we used His-tagged p32 as the bait instead of the untagged p32 utilized in the other pulldowns. We apologize for failing to include the details of which His-tagged bait protein was used in each lane; we have updated the figure and figure legend as shown below, which we believe addresses this issue. We hope that these changes have improved the interpretability of our pulldown results (Fig. 5f).

Response letter_Fig. 1 His-tag pull-down of purified human proteins *in situ* demonstrates direct interaction between C1q-BAI1, C1q-p32, and BAI1-p32. Bait red, Western blot antibodies blue. Blots have been cropped and stitched together for better visualization (grey line between the lanes). The full original blots are in the Source data.

In addition, we have included the details in the Methods section, page 19, row 11 as follows:

“Pull-down protein interaction analysis

Direct interaction between C1q, BAI1 and p32 *in situ* was tested by pulldown assays using purified human proteins following the manufacturer instructions. Briefly, 5 μ g His-tag recombinant human BAI1 (SinoBiological, 4969-BA) or His-tag p32 (SinoBiological, 11874-H08E) were prepared as bait in a 1x binding/wash buffer (50 mM sodium phosphate, 300 mM NaCl, 0.01% Tween-20) and immobilized on 2 mg magnetic beads (Dynabeads His-Tag Isolation and Pulldown, Invitrogen, 10103D) by incubation on a rocker for 10 min at RT. Tubes containing the bait protein immobilized on beads were placed on a magnet (Invitrogen, 12321D) and the supernatant containing excess bait protein was collected for Western blot validation. Beads were washed four times with 1x binding/wash buffer to remove any remaining excess bait protein. 5 μ g of prey proteins purified human C1q (MyBioSource, MBS147305) and recombinant untagged human p32 (LSBio, LS-G3375-20) were prepared in a 1x pulldown buffer (3.25 mM sodium phosphate, 70 mM NaCl, 0.01% Tween-20) and incubated with the immobilized bait protein on a rocker for 15 min at RT. Prior to pulldown protein capture, tubes containing the immobilized bait-prey protein complexes were placed on a magnet and washed four times with 1x binding/wash buffer to remove any remaining unbound prey proteins. Bait protein and bound prey proteins were eluted using his-elution buffer (300 mM imidazole, 50 mM sodium phosphate, 300 mM NaCl, 0.005% Tween-20) and the eluted samples were analyzed using Western blotting. The original blots are in the Source data.”

2) Related to the above, the authors indicate in a number of instances in the response that "The full original blots are in the Source data." This reviewer looked through every file in the manuscript tracking system and could not locate any original blots. These blots would help to interpret the data mentioned in point #1 above.

We apologize for the confusion. We were not able to upload the source file (an excel file) directly due to its complexity and size into the Nature Communications portal (only pdf allowed). The document was uploaded for us by the submission support and we were not able to validate the document afterwards. Our best guess is that something happened and the photos of the western blot membranes for Fig. 1m, Fig. 2b, Fig. 3b, Fig. 3i, Fig. 4r, Fig. 4s, Fig. 4y, Fig. 5e, Fig. 5f, and Supplementary Fig. 3a that we had

attached into the corresponding sheets were lost during this process. The source data file with the photos of the western blot membranes is included into this resubmission again.

Reviewer #4 (Remarks to the Author):

We thank Reviewer for their time co-reviewing our manuscript.